# Reader domain specificity and lysine demethylase-4 family function

Zhangli Su[1,2,*], Fengbin Wang[3,*], Jin-Hee Lee[1,2], Kimberly E. Stephens[4,5], Romeo Papazyan[4,5,†], Ekaterina Voronina[6], Kimberly A. Krautkramer[1,2], Ana Raman[4,5], Jeremy J. Thorpe[4,5], Melissa D. Boersma[1,2], Vyacheslav I. Kuznetsov[1,2], Mitchell D. Miller[3], Sean D. Taverna[4,5], George N. Phillips Jr[3,7,8] & John M. Denu[1,2]

The KDM4 histone demethylases are conserved epigenetic regulators linked to development, spermatogenesis and tumorigenesis. However, how the KDM4 family targets specific chromatin regions is largely unknown. Here, an extensive histone peptide microarray analysis uncovers trimethyl-lysine histone-binding preferences among the closely related KDM4 double tudor domains (DTDs). KDM4A/B DTDs bind strongly to H3K23me3, a poorly understood histone modification recently shown to be enriched in meiotic chromatin of ciliates and nematodes. The 2.28 Å co-crystal structure of KDM4A-DTD in complex with H3K23me3 peptide reveals key intermolecular interactions for H3K23me3 recognition. Furthermore, analysis of the 2.56 Å KDM4B-DTD crystal structure pinpoints the underlying residues required for exclusive H3K23me3 specificity, an interaction supported by *in vivo* co-localization of KDM4B and H3K23me3 at heterochromatin in mammalian meiotic and newly postmeiotic spermatocytes. *In vitro* demethylation assays suggest H3K23me3 binding by KDM4B stimulates H3K36 demethylation. Together, these results provide a possible mechanism whereby H3K23me3-binding by KDM4B directs localized H3K36 demethylation during meiosis and spermatogenesis.

[1] Wisconsin Institute for Discovery, Morgridge Institute for Research, University of Wisconsin–Madison, Madison, Wisconsin 53715, USA. [2] Department of Biomolecular Chemistry, School of Medicine and Public Health, University of Wisconsin–Madison, 330 North Orchard Street, Madison, Wisconsin 53715, USA. [3] Biosciences at Rice, Rice University, Houston, Texas 77005, USA. [4] Department of Pharmacology and Molecular Sciences, The Johns Hopkins University School of Medicine, Baltimore, Maryland 21205, USA. [5] Center for Epigenetics, The Johns Hopkins University School of Medicine, Baltimore, Maryland 21205, USA. [6] Division of Biological Sciences, University of Montana, Missoula, Montana 59812, USA. [7] Department of Chemistry, Rice University, Houston, Texas 77005, USA. [8] Department of Biochemistry, University of Wisconsin–Madison, Madison, Wisconsin 53715, USA. * These authors contributed equally to this work. † Present address: Division of Endocrinology, Diabetes and Metabolism, Department of Medicine, Perelman School of Medicine, University of Pennsylvania, Philadelphia, Pennsylvania 19104, USA. Correspondence and requests for materials should be addressed to J.M.D. (email: jmdenu@wisc.edu), S.D.T. (email: staverna@jhmi.edu) or to G.N.P. (email: georgep@rice.edu).

Histone lysine methylation regulates gene expression by recruiting or displacing chromatin-binding proteins[1–4]. KDM4 (JMJD2) is a conserved iron (II)-dependent jumonji-domain demethylase subfamily that is essential during development[5–8]. Disrupting the only KDM4 enzyme in *Caenorhabditis elegans* induced germ cell apoptosis and DNA replication defects[9]. Overexpression of individual mammalian KDM4 proteins has been associated with oncogenesis, cancer growth and metastasis in various cancer types and other conditions including cardiac failure and autism[10–12]. In vertebrates, KDM4A, KDM4B and KDM4C share similar domain organization[13] (Fig. 1a). The amino-terminal catalytic domains of KDM4A–C display demethylase activity that can convert di-/trimethylated lysines to lower methylated states at H3K9 and H3K36 with comparable kinetics[13]. Despite similar catalytic activities, individual KDM4 members exhibit varied chromatin associations and biological functions[14–16]. These observations suggest an uncharacterized mechanism controls KDM4 protein functions on chromatin.

Vertebrate KDM4A–C proteins contain a conserved double tudor domain (DTD) and a potential zinc-finger domain at the carboxy terminus (Fig. 1a). These types of chromatin-interacting modules (also known as reader domains) often mediate binding to specific histone modification states[17–19]. Tudor domains are part of the 'Royal Family' reader domains, which usually recognize methylated lysine residues[20,21]. In particular, DTD from KDM4A (KDM4A-DTD) was shown to form an unique integral structural unit and recognize methylated lysines[22–27]. Deletion of the C-terminal domain in KDM4 proteins resulted in a change of sub-cellular localization, changed demethylase activity and disruption of other KDM4 functions[8,14,15,28], suggesting functional roles for the C-terminal DTDs. However, there has been no comprehensive investigation of the histone-binding properties for KDM4B and KDM4C DTDs.

To better understand how reader domains regulate the overall chromatin-acting functions among the closely related KDM4 family members, we aimed to determine and compare histone interactomes of the C-terminal DTDs in human KDM4A–C proteins (Fig. 1a). From our biochemical and structural profiling, we find KDM4A, KDM4B and KDM4C DTDs display different histone-binding preferences. We show that these DTDs use an aromatic cage as a general mechanism to coordinate trimethyl lysine and, most importantly, the sequence specificity is largely determined by side-chain interactions with surrounding residues. Specifically, we describe the unique interaction between KDM4-DTDs and H3K23me3, a histone modification enriched in heterochromatin during meiosis in primary spermatocytes. Our crystal structures and homology models explain the origins of H3K23me3 specificity by KDM4B, and these biochemical and structural data are supported by the co-localization of full-length KDM4B with H3K23me3 *in vivo*. We demonstrate that H3K23me3 binding stimulates H3K36 demethylation by KDM4B. These results suggest H3K23me3 as a novel link between KDM4 family and mammalian germ cell development, potentially to eliminate methylation at H3K36 at heterochromatic regions. The molecular determinants of histone-binding specificity by DTDs serves as a general framework to understand KDM4 family functions.

## Results

### H3K23me3-binding specificities among KDM4 DTDs.

We cloned and expressed human KDM4A–C DTDs as recombinant proteins with N-terminal affinity tags from *Escherichia coli* (Supplementary Fig. 1a). KDM4A–C DTDs were probed with our recently developed combinatorial histone peptide microarray covering 746 histone post-translational modification (PTM) states[29] (Supplementary Fig. 1b). Analysis of the peptide microarray assay revealed surprising discrimination for H3K23me3 binding among the three DTDs (Fig. 1b). In particular, KDM4B-DTD displayed exclusive binding to H3K23me3 (Fig. 1b). KDM4A-DTD, which displayed strong binding to H3K23me3 as well, also bound H3K4me3 and H4K20me3 (Fig. 1b), consistent with previous findings[23,30]. In contrast, KDM4C-DTD bound specifically to H3K4me3 (Fig. 1b).

We further quantified the methylated histone binding of KDM4-DTDs by employing a solution-based binding assay (fluorescence polarization; Fig. 1c,d). Overall, the derived binding constants were consistent with the peptide array analysis. KDM4A-DTD binds H3K23me3 peptide at low micromolar affinity ($K_d = 2.2 \mu M$), slightly stronger than its interaction with H3K4me3 and H4K20me3 (Fig. 1d), and KDM4C-DTD displayed highly specific binding to H3K4me3 ($K_d = 6.8 \mu M$; Fig. 1d). In dramatic contrast to KDM4A-DTD and KDM4C-DTD, KDM4B-DTD only displayed striking specificity for H3K23me3 ($K_d = 10.3 \mu M$), with no significant binding to H3K4me3 (Fig. 1d and Supplementary Fig. 1c). Collectively, this systematic analysis indicates that KDM4A–C DTDs have distinct and surprising histone-binding preferences that could not have been predicted based on the high similarity in their overall amino acid sequences ($>60\%$ identity, $>80\%$ similarity; see Fig. 1e). Nevertheless, how KDM4-DTDs achieve sequence specificity for residues proximal to the trimethyl lysine remained unclear. To define the mechanism of such sequence selectivity for KDM4-DTD binding, we proceeded to structurally and biochemically detail these interactions.

### Structural basis for H3K23me3 recognition.

To elucidate the molecular basis of recognizing H3K23me3, purified KDM4A-DTD was co-crystallized with an H3 (19–28) K23me3 peptide and the peptide-bound structure was determined at 2.3 Å with a space group of *P 32* (PDB ID: 5D6Y; Fig. 2a–c, Table 1 and Supplementary Fig. 2a). From the crystallization screening, we also determined two apo structures of KDM4A-DTD at 1.99 and 2.15 Å (PDB ID: 5D6W, 5D6X; Table 1). In all the structures, KDM4A-DTD forms two lobes (HTD-1 and HTD-2; HTD, hybrid tudor domain) with interweaving β-strands. The co-crystal structure suggests the H3K23me3 peptide forms extensive interaction with KDM4A-DTD ($\sim 540 Å^2$ by PISA[31]) and such binding is mostly mediated through residues in HTD-2 of KDM4A-DTD (Figs 1e and 2a).

The structure of the H3K23me3-bound KDM4A-DTD shows that the trimethyl lysine group of H3K23 is coordinated via cation-π and/or hydrophobic interactions by an aromatic cage formed by F932, W967 and Y973 of the reader domain (Fig. 2d). To examine the selectivity of methylation state on H3K23, we determined binding constants of KDM4A-DTD for mono-/di-/trimethylated H3K23 peptides. The binding results suggest the KDM4A-DTD aromatic cage favours trimethyl lysine ($K_d = 2.20 \mu M$) over dimethyl lysine ($K_d = 9.06 \mu M$), or monomethyl lysine ($K_d = 98.55 \mu M$) at H3K23 (Fig. 2e). Correspondingly, H3K23me3 binding is greatly decreased ($K_d = 284.90 \mu M$) with the aromatic cage substitution Y973A (Fig. 2e). The conserved role of this aromatic cage in directing PTM selectivity (Supplementary Fig. 2b) towards trimethyl lysine suggests specific recognition with H3K23me3 is influenced by neighbouring residues besides the trimethyl lysine moiety on histones. The orientation of H3K23me3 peptide in KDM4A is more similar to H3K4me3 than H4K20me3 peptide (Supplementary Fig. 2b). However, other H3K4me3 reader domains such as ING2-PHD

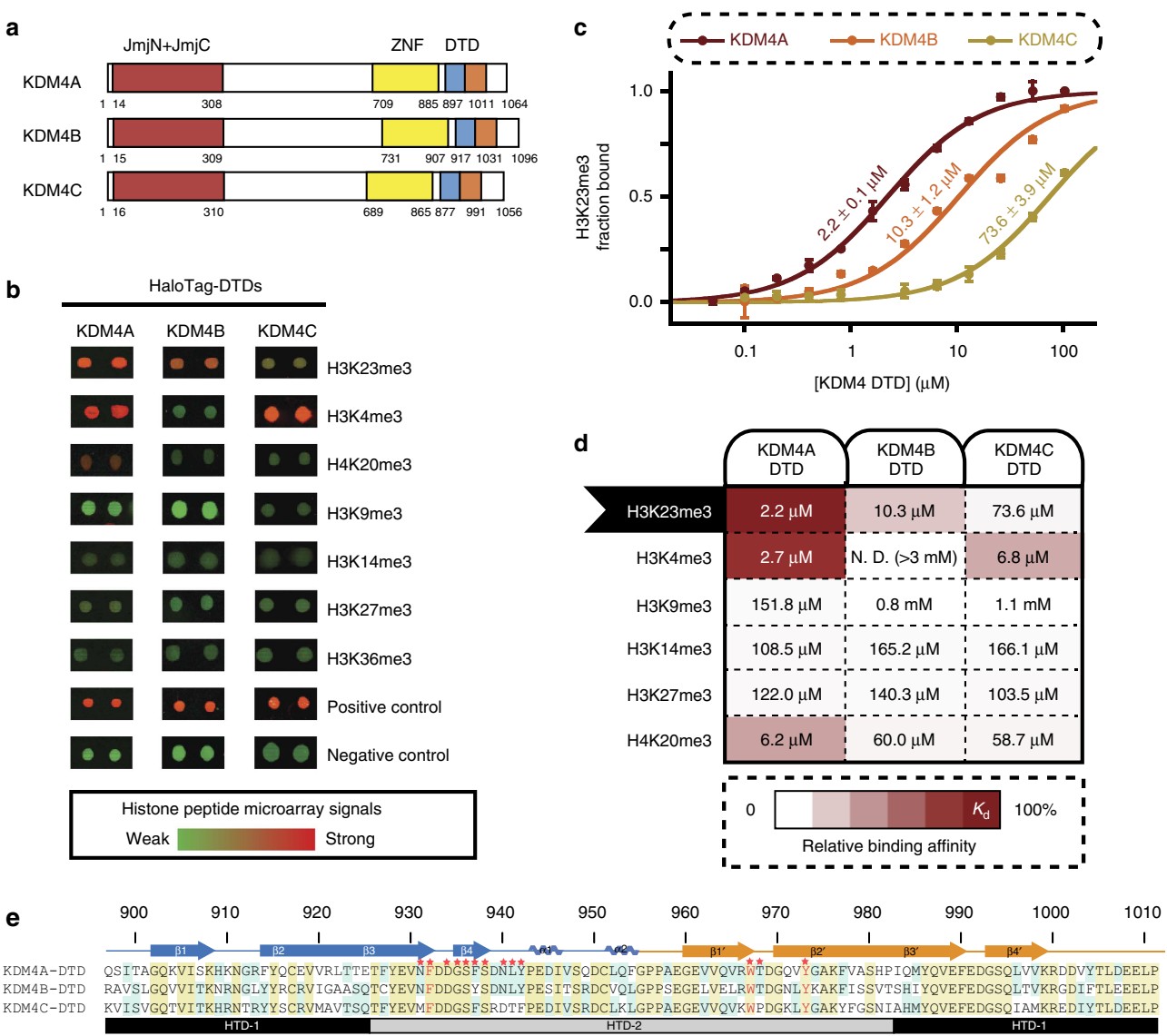

**Figure 1 | Distinct binding specificities of human KDM4A–C DTDs.** (**a**) Domain organization of human KDM4 family members (DTD, double tudor domain; JmjN + JmjC, jumonji catalytic demethylase domain; ZNF, zinc finger). DTDs were coloured in two blocks (blue and orange) with each representing a tudor domain sequence. (**b**) Systematic profiling of histone binding preferences of KDM4A–C DTDs on histone peptide microarray. Recombinant HaloTag-tagged DTDs were expressed and purified form *E. coli* and incubated with in-house histone peptide microarray. Individual spots were selected from the representative array images (see Supplementary Fig. 1 for the full images). Red signals correlate with binding. (**c**) Quantitative measurement of H3 (17–32) K23me3 binding constant with KDM4A–C DTDs by fluorescence polarization (FP). Errors represent s.d. from three experimental replicates. (**d**) Differential histone interactome of KDM4A–C DTDs validated by FP assays. Matrix represents pairs of KDM4 DTD proteins and trimethyl-lysine peptides (see Supplementary Table 1 for peptide information) for each FP assay. Relative specificity factors (as shown by the shades of red) were normalized to the tightest binding in the matrix ($K_d = 2.2 \mu M$). Binding affinity for each protein-peptide combination as determined by FP is labelled. (**e**) Sequence alignment of KDM4A–C DTDs. Secondary structure is labelled on top of the sequence (blue: Tudor 1, orange: Tudor 2). Shades indicate conservation of particular residues (yellow: conserved in all three proteins; cyan: conserved in two proteins). Red stars highlight the residues that interact with H3K23me3 peptide as determined from Fig. 2. Letters in red represent aromatic residues.

could not bind H3K23me3 (Supplementary Fig. 2c), because ING2-PHD recognizes the free N terminus (H3A1) of H3K4me3 peptide and does not accommodate the longer N terminus of H3K23me3. In contrast, the more open channel between the two lobes in DTDs can accommodate the longer polypeptide chain N-terminal to H3K23me3.

H3T22, the neighbouring residue of H3K23, plays an important role in the H3K23me3 recognition by KDM4A-DTD. H3T22 side chain forms a hydrogen bond with N940, whereas its main chain forms hydrogen bonds with N940 and S938 side chains (Fig. 2f). The importance of H3T22 in H3K23me3-binding

is illustrated by the decreased H3K23me3 binding with phosphorylation on H3T22 ($K_d = 83.69 \mu M$) or when N940 is changed into Ala in KDM4A ($K_d = 57.41 \mu M$; Fig. 2g), both of which would interfere with the hydrogen bond of H3T22 side chain. The effect of H3T22ph on H3K23me3 binding is reminiscent of the relationship between H3T3ph and H3K4me3. Indeed, a similar loss of KDM4A binding to H3K4me3 was observed with H3T3ph (Supplementary Fig. 2d).

Unique contacts with H3R26 also contribute to H3K23me3 recognition by KDM4A. The H3R26 side chain forms a hydrogen bond with KDM4A N931 (2.9 Å), whereas the main chain of

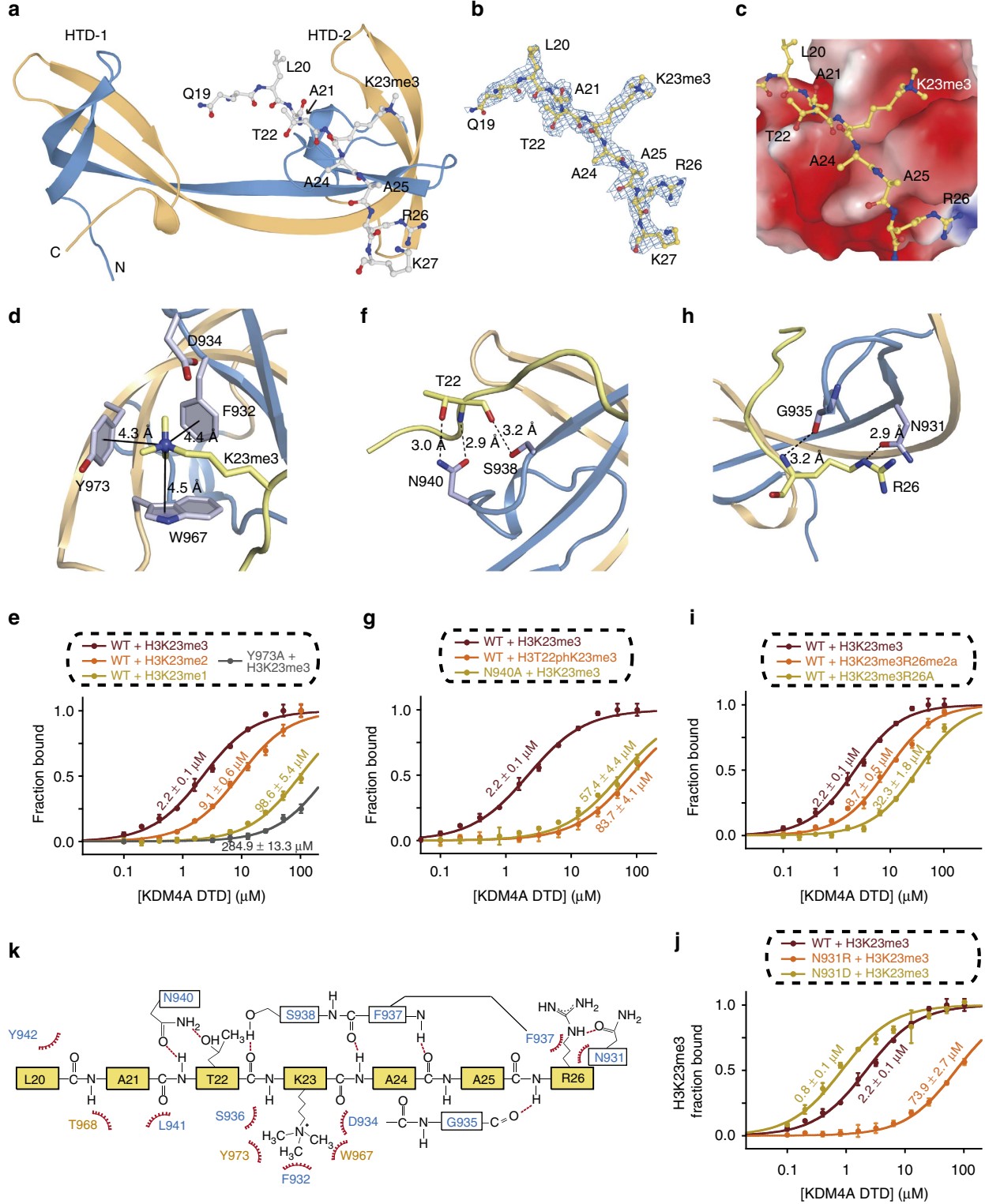

**Figure 2 | Co-crystal structure of KDM4A-DTD with H3K23me3.** (a–c) Co-crystal structure of KDM4A-DTD and H3 (19–27) K23me3 peptide at 2.28 Å resolution (PDB ID: 5D6Y). Data collection and refinement statistics are summarized in Table 1. (d–j) Biochemical validation of intermolecular interactions observed in KDM4A-H3K23me3 structure. Aromatic cage (d,e), H3T22-N940 pair (f,g) and H3R26–N931 pair (h–j) contribute to KDM4A-H3K23 recognition. Binding affinity of corresponding protein-peptide combinations is quantified by fluorescence polarization assays. Errors represent s.d. from three experimental replicates. See Supplementary Table 1 for peptide information. (k) Schematic representation of the molecular interactions between KDM4A-DTD and H3(20–26)K23me3 peptide as identified in the co-crystal structure (PDB ID: 5D6Y). Colouring of KDM4A-DTD in a,d,f,h and k corresponds to Fig. 1e.

**Table 1 | Data collection and refinement statistics.**

| | KDM4B-DTD apo form | KDM4A-DTD H3K23me3 | KDM4A-DTD apo form 1 | KDM4A-DTD apo form 2 |
|---|---|---|---|---|
| *Data collection* | | | | |
| Space group | *P* 3₁ | *P* 3₂ | *R* 32:*h* | *I* 2₁3 |
| Cell dimensions | | | | |
| *a, b, c* (Å) | 77.4, 77.4, 50.8 | 106.2, 106.2, 79.2 | 190.6, 190.6, 117.0 | 134.8, 134.8, 134.8 |
| *α, β, γ* (°) | 90.0, 90.0, 120.0 | 90.0, 90.0, 120.0 | 90.0, 90.0, 120.0 | 90.0, 90.0, 90.0 |
| Resolution (Å) | 30–2.56 (2.68–2.56) | 30–2.28 (2.32–2.28) | 36–1.99 (2.01–1.99) | 42–2.15 (2.28–2.15) |
| $R_{merge}$ | 0.120 (0.797) | 0.225 (1.885) | 0.130 (1.158) | 0.076 (0.523) |
| $I/\sigma I$ | 6.73 (2.28) | 12.95 (1.42) | 12.6 (2.35) | 25.1 (5.74) |
| Completeness (%) | 100.0 (100.0) | 100.0 (100.0) | 100.0 (100.0) | 100.0 (100.0) |
| Redundancy | 6.6 (6.3) | 11.3 (8.7) | 9.8 (9.7) | 13.8 (13.8) |
| $CC^{1/2}$ | 0.99 (0.74) | 0.99 (0.46) | 0.99 (0.67) | 0.99 (0.95) |
| | | | | |
| *Refinement* | | | | |
| Resolution (Å) | 30–2.56 | 30–2.28 | 36–1.99 | 42–2.15 |
| No. reflections | 71,716 | 466,599 | 542,229 | 306,479 |
| $R_{work}/R_{free}$ | 0.197/0.254 | 0.264/0.295 | 0.168/0.195 | 0.159/0.199 |
| Peptide RSCC* | — | 0.92 | — | — |
| No. atoms (non-H) | | | | |
| Protein | 1,716 | 5,069 | 3,795 | 1,974 |
| Peptide | — | 248 | — | — |
| Water | 66 | 216 | 515 | 242 |
| *B*-factors | | | | |
| Protein | 48.2 | 42.0†/99.1‡ | 39.6 | 34.9 |
| Peptide | — | 46.0 | — | — |
| Water | 44.7 | 48.3 | 53.0 | 50.2 |
| Root mean squared deviations | | | | |
| Bond lengths (Å) | 0.009 | 0.011 | 0.008 | 0.008 |
| Bond angles (°) | 1.12 | 1.39 | 1.10 | 1.11 |
| PDB code | 4UC4 | 5D6Y | 5D6W | 5D6X |

*RSCC is the real-space correlation to electron density calculated by phenix.
†Average B-factor of ordered chains A–D in asymmetric unit.
‡Average B-factor of disordered chains E–F in asymmetric unit.

H3R26 forms a hydrogen bond with the KDM4A-DTD G935 main chain (Fig. 2h). H3R26 fits nicely into a negatively charged cleft formed by residues spanning from E929 to D933 (Supplementary Fig. 2e). To provide direct biochemical evidence of the H3R26–N931 interaction pair in H3K23me3 recognition, we modified H3R26 in the context of H3K23me3 peptide and observed decreased binding (32.3 μM) between wild-type KDM4A-DTD and H3K23me3R26A peptide (Fig. 2i). Interestingly, modifying H3R26 with asymmetric dimethylation dropped H3K23me3 binding mildly to 8.7 μM (Fig. 2i), suggesting KDM4A-DTD might read H3K23me3 and H3R26me2a in a combinatorial manner. We also observed a dramatic decrease in H3K23me3 binding (73.9 μM) with N931R substitution (Fig. 2j), which could form electrostatic repulsion with H3R26. Conversely, changing N931 into aspartic acid (N931D) increased H3K23me3 binding by about threefold ($K_d = 0.85$ μM; Fig. 2j), potentially by providing ionic interaction with H3R26. Interestingly, N931D also increased H3K14me3 binding ($K_d = 12.24$ μM; Supplementary Fig. 2f), likely to be due to the enhanced ionic interaction between D931 and H3R17 at the +3 site of H3K14me3, reminiscent of the distance between H3K23me3 and H3R26 at the +3 position (Supplementary Fig. 2g). Overall, the H3R26–N931 pair mediates H3K23me3 recognition by KDM4A-DTD and this contact is unique to H3K23me3 binding. Besides main-chain hydrogen bonds between H3R26 and G935, other main-chain hydrogen bonds are found between H3A24 and F937, together forming three β-sheet-like main-chain hydrogen bonds. The intermolecular interactions determined from the KDM4A-H3K23me3 co-crystal structure involve all residues from H3L20 to H3R26 (Fig. 2k), consistent with the clear electron density of this region in the structure (Fig. 2b). The

molecular arrangement of the H3K23me3-bound structure defines an H3K23me3-specific interaction network and suggests that modifications on H3T22 and H3R26 could play regulatory roles in modulating DTD binding.

**H3K23me3 recognition by specific side-chain interactions.** Previously, H3K4me3 and H4K20me3 were characterized as histone PTMs recognized by KDM4A-DTD[23,30]. D945 of KDM4A was identified as the residue important for H3K4me3 binding by forming a salt bridge with H3R2, whereas D939 was reported critical for H4K20me3 binding due to its interaction with H4R19. We noted that neither D945 nor D939 contacts the H3K23me3 peptide in the KDM4A–H3K23me3 structure (Fig. 2k). Using this observation, we generated a series of D945 substitutions (D945A/L/S) and found that they did not affect H3K23me3 binding, although such changes did decrease H3K4me3 binding (Supplementary Fig. 3a,b). Similarly, D939A only weakens H4K20me3 binding but not H3K23me3 binding or H3K4me3 binding (Supplementary Fig. 3a–c). The fact that individual KDM4-DTD substitutions at different residues could exclusively disrupt H3K23me3, H3K4me3 and H4K20me3 binding, respectively, without affecting the other two interactions, demonstrates that H3K23me3 recognition is distinct from either H3K4me3 or H4K20me3 interactions.

To further confirm the specific H3K23me3 recognition is mediated by unique side chain interactions, we combined amino acid substitutions that reduce H3K4me3 (D945R) and H4K20me3 (D939R) affinity, while increasing H3K23me3 affinity (N931D) as a triple mutant ('DRR') in KDM4A-DTD (Fig. 3a,b). The DRR mutant retained a low micromolar $K_d$ value for

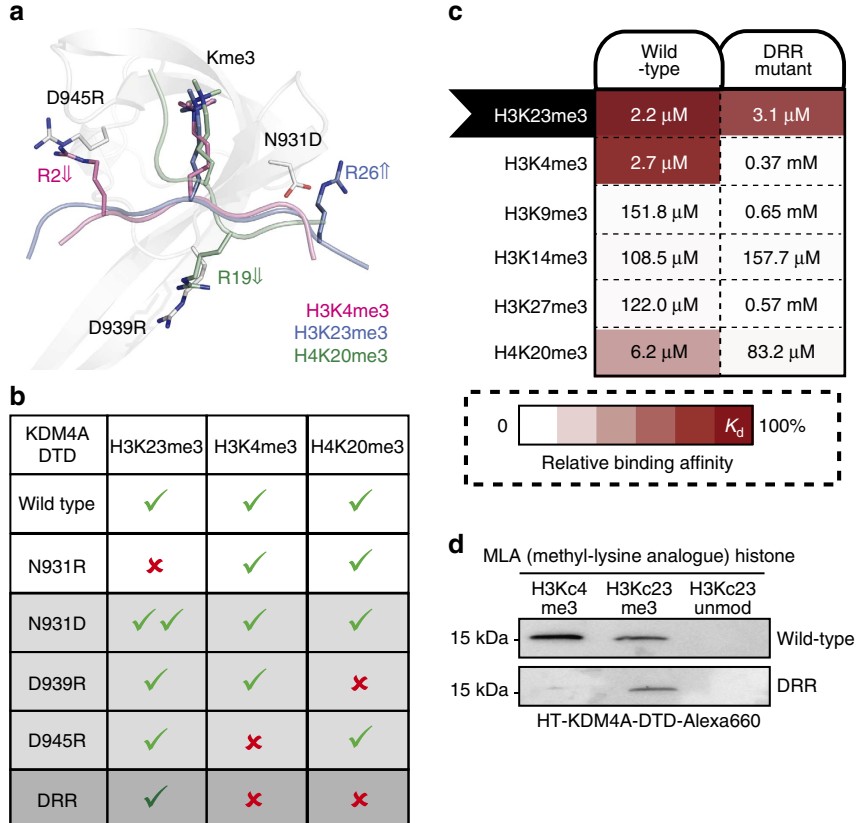

**Figure 3 | H3K23me3 recognition is modulated by unique side-chain interactions.** (**a,b**) Structural representation and a table summary about the side-chain interactions for trimethyl-lysine histone recognition by KDM4A-DTD: N931D increases H3R26-mediated H3K23me3 interaction, D939R decreases H4R19-mediated H4K20me3 interaction and D945R decreases H3R2-mediated H3K4me3 interaction. (**c**) Triple mutant (N931D/D939R/D945R, or DRR) reader is specific for H3K23me3. Matrix represents pairs of KDM4 DTD proteins and trimethyl-lysine peptides (see Supplementary Table 1 for peptide information) for each fluorescence polarization assay. Relative specificity factors (as shown by the shades of red) were normalized to the tightest binding in the matrix ($K_d = 2.2 \mu M$). Binding affinity for each protein-peptide combination as determined by FP is labelled. (**d**) DRR mutant fails to recognize H3Kc4me3 MLA but retains binding to H3Kc23me3 MLA histone in a far western blot setting.

H3K23me3 binding, while discriminating against other PTM states, leading to overall enhanced H3K23me3 specificity (Fig. 3c). The engineered 'DRR' reader domain specifically recognized H3K23me3 full-length recombinant histone (Fig. 3d). The successful engineering to selectively recognize H3K23me3-containing histones highlights the detailed molecular knowledge gleaned from this study.

**KDM4B-DTD is an H3K23me3-specific reader domain.** The above biochemical analysis validated the intermolecular interactions between KDM4A-DTD and H3K23me3 observed from the co-crystal structure. As KDM4B-DTD also interacts with H3K23me3 with high discrimination (Fig. 1d), we investigated whether KDM4B-DTD engages H3K23me3 using intermolecular interactions similar to KDM4A-DTD. Indeed, KDM4B-DTD binds H3K23me3 with sensitivity to PTM states on H3K23 and neighbouring residues in a similar fashion to that observed for KDM4A-DTD (Fig. 4a,b), suggesting that the interaction network with H3K23me3 is shared between KDM4A-DTD and KDM4B-DTD. However, our initial analysis showed a unique specificity of KDM4B-DTD: unlike KDM4A-DTD and KDM4C-DTD, KDM4B-DTD displayed dramatically poor interaction with H3K4me3 (Fig. 1b,d and Supplementary Fig. 1c). Such poor interaction with H3K4me3 was also confirmed with more physiologically relevant chromatin substrates, native nucleosomes purified from human cell line MCF-7, and KDM4B-

DTD showed no enrichment of H3K4me3-containing chromatin (Fig. 4c). Furthermore, KDM4B-DTD displayed H3K23me3-specific binding with reconstituted methyl lysine analogue (MLA) nucleosomes and such binding was abolished with the aromatic cage mutant Y993A (Fig. 4d). To better understand the molecular basis of the specific H3K23me3 binding and the disfavoured H3K4me3 binding by KDM4B-DTD, we crystallized KDM4B-DTD and successfully determined the structure of the apo-form at a resolution of 2.56 Å (PDB ID: 4UC4; Table 1). Analysis of the KDM4B-DTD structure reveals a highly similar overall tertiary structure when compared with KDM4A-DTD and KDM4C-DTD (Supplementary Fig. 4a). Next, we modelled KDM4B-DTD with the H3(19–27)K23me3 (Fig. 4e) or H3(1–7)K4me3 (Fig. 4f–i) peptide, based on the highly conserved structural fold of HTD-2 from the corresponding KDM4A co-crystal structures (PDB ID: 5D6Y and 2GFA). Indeed, the KDM4B-H3K23me3 model displayed a similar interaction network (Supplementary Fig. 4b,c) as KDM4A-H3K23me3 (Fig. 2k), including the hydrogen bonds with H3T22 and H3R26. Specific KDM4B mutants that impaired its H3K23me3 binding also confirmed the KDM4B-H3K23me3 model (Supplementary Fig. 4d). Furthermore, wild-type KDM4B-DTD but not the aromatic cage mutant (Y993A) displayed binding to recombinant H3K23me3 histone (Supplementary Fig. 4e).

To pinpoint the residues that account for the poor H3K4me3 affinity of KDM4B, we looked for apparent clashes or missing molecular interactions in the modelled KDM4B-H3K4me3

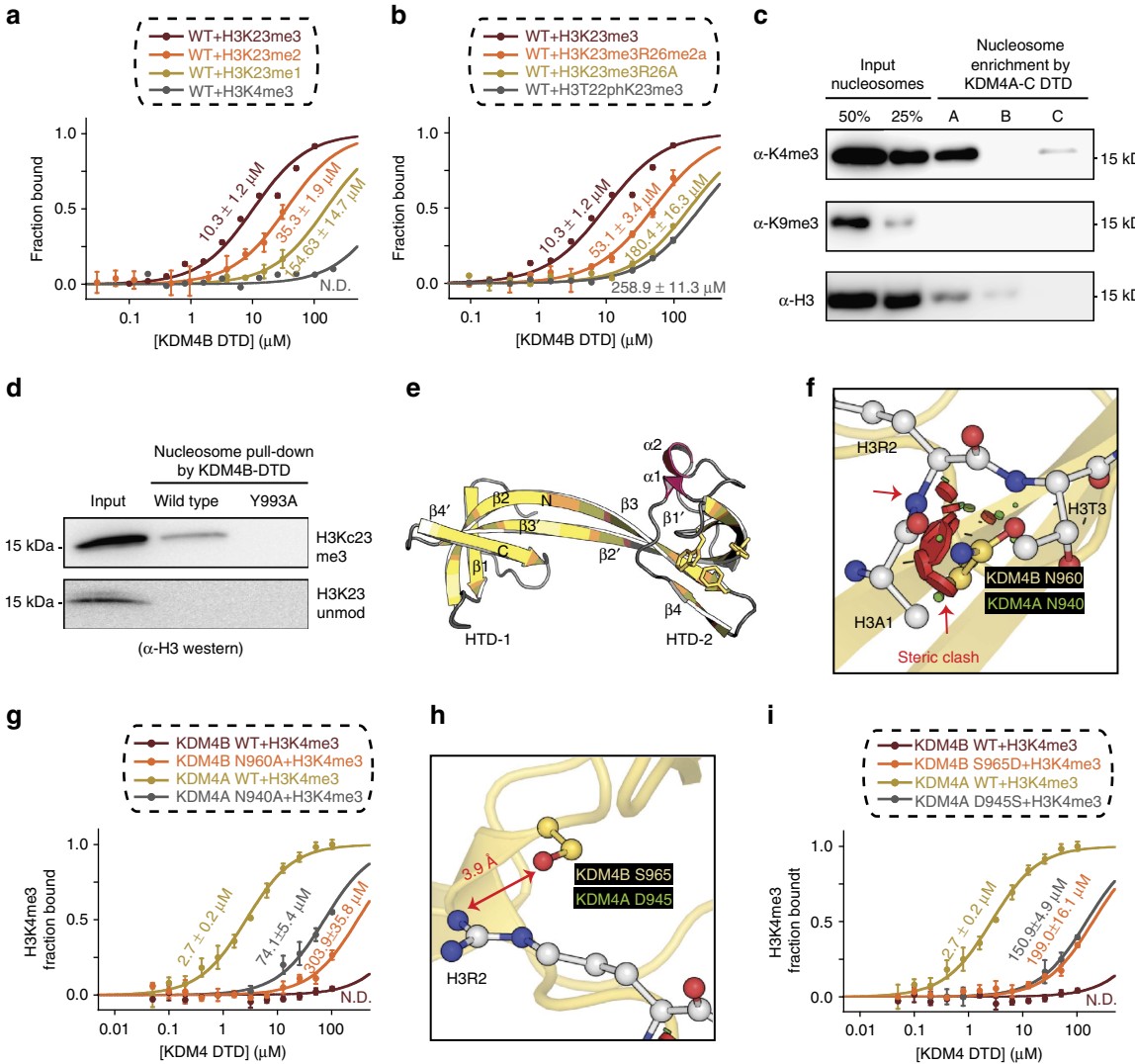

**Figure 4 | KDM4B-DTD is an H3K23me3-specific reader.** (**a,b**) Conserved interaction between KDM4B-DTD and H3K23me3 as determined by FP. (**c**) MARCC confirms enrichment of H3K4me3-containing chromatin by KDM4A and KDM4C but not KDM4B. HaloTag-tagged DTDs are immobilized on resin and incubated with native mononucleosomes purified from MCF-7 cells. Enriched nucleosomes are eluted with SDS sample buffer and probed by indicated antibodies. (**d**) Recombinant KDM4B-DTD wild type pulls down H3Kc23me3-containing MLA nucleosome (H3Kc23me3Kc36me3). Aromatic cage mutant KDM4B-DTD Y993A and H3K23 unmodified-containing MLA nucleosome (H3K23unmodKc36me3) were included as controls. (**e**) Based on the structural alignment of HTD-2 domains, H3(19–27)K23me3 peptide from KDM4A structures is modelled into KDM4B-DTD apo structure (PDB ID: 4UC4). See Methods for details. (**f–i**) Key residues contributing to decreased H3K4me3 binding identified from KDM4B-H3K4me3 model. Steric clash (red disk: major clash) between KDM4B N960 and H3A1/R2 (**f,g**). Decreased interaction between KDM4B S965 and H3R2 (**h,i**). Corresponding residues in KDM4A are labelled. Enhanced H3K4me3 binding by KDM4B mutants N960A and S965D. As controls, corresponding mutants in KDM4A are examined for H3K4me3 binding. (**a,b,g,i**) Binding affinity of corresponding protein-peptide combinations is quantified by FP. Errors represent s.d. from three experimental replicates.

structure (Fig. 4f–i). We first noticed that N960 in KDM4B-DTD showed a significant steric clash with H3A1 and H3R2 in the model (Fig. 4f). To test whether N960 hinders the interaction between H3K4me3 and KDM4B-DTD, we made the Asn-to-Ala substitution, which should alleviate the steric clash. Indeed, with the N960A variant, H3K4me3 binding was rescued to $K_d = 303\,\mu M$, whereas the corresponding substitutions in KDM4A (N940A) dramatically decreased H3K4me3 binding (Fig. 4g), most probably due to disrupting the hydrogen bond with H3T3. S965 in KDM4B might also contribute to the poor affinity for H3K4me3, probably due to the lack of strong ionic interaction between H3R2 and S965; such ionic interaction is fulfilled between KDM4A-D945 and H3R2 (Fig. 4h). In addition, KDM4B-S965 forms a weaker hydrogen bond (3.9 Å) with

H3R2 in the modelled structure than KDM4A-D945 (3.0 Å; Fig. 4h). Consistent with this hypothesis, H3K4me3 binding in KDM4B-S965D ($K_d \sim 200\,\mu M$) is tighter than that in wild-type KDM4B ($K_d$ not measurable; Fig. 4i). At the same time, corresponding substitutions of KDM4A-D945S displayed lower H3K4me3 binding than the corresponding wild-type proteins (Fig. 4i). Similar models of KDM4C-DTD with H3K4me3 and H3K23me3 suggests general conserved interactions as with KDM4A-DTD, except for the H3R26 clash in KDM4C-H3K23me3 model (Supplementary Fig. 4f,g), which might explain KDM4C's lower affinity to H3K23me3. Together, these results suggest that despite overall sequence and structure homology, specific differences of key residues result in unique H3K23me3 specificity of KDM4B-DTD. This also highlights the

multiple side chain interactions that define the sequence specificity of KDM4 reader domains.

**H3K23 methylation is enriched in spermatocytes.** The specific H3K23me3 interaction by human KDM4 reader domains *in vitro* prompted us to investigate further the physiological context of such interaction. Only recently has a biological context for the histone PTM H3K23me3 been described in *Tetrahymena* and *C. elegans*[32,33]. However, direct evidence of its relevance in mammals has not been documented. To search for a physiologically relevant source of H3K23me3 in mammalian systems, we first acid-extracted endogenous histones from a panel of mammalian cell lines and mouse tissues (Supplementary Fig. 5a). We also purified histones from germline micronuclei and non-germline macronuclei in *Tetrahymena*, as H3K23me3 was previously found exclusively enriched in the micronuclei[32]. Using an H3K23me3-specific antibody produced in-house[32] and

profiled on peptide microarray (Supplementary Fig. 5b,c), we found specific enrichment of H3K23me3 in mouse testes compared with other non-germline sources (Fig. 5a). Higher H3K23me3 levels in germline (micronuclei) histone were detected in *Tetrahymena* (Fig. 5a), consistent with previous observations. Using the recombinant H3K23me3-specific reader as a probe (DRR mutant), a similar enrichment pattern was observed in testes histones (Fig. 5b).

To test whether H3K23me3 detected in mouse testes also displays association with meiotic chromatin as found previously in *Tetrahymena* and *C. elegans*[32], we performed indirect immunofluorescence of H3K23me3 in mouse testes (Fig. 5c and Supplementary Fig. 5d). We detected an enrichment of H3K23me3 in mouse primary spermatocytes as they began meiosis I, as indicated by the appearance of the meiosis marker SCP3 (synaptonemal complex protein 3), which coincides with meiotic entry and leptotene. In new spermatids, which had completed meiosis II, H3K23me3 was also detected, albeit in a

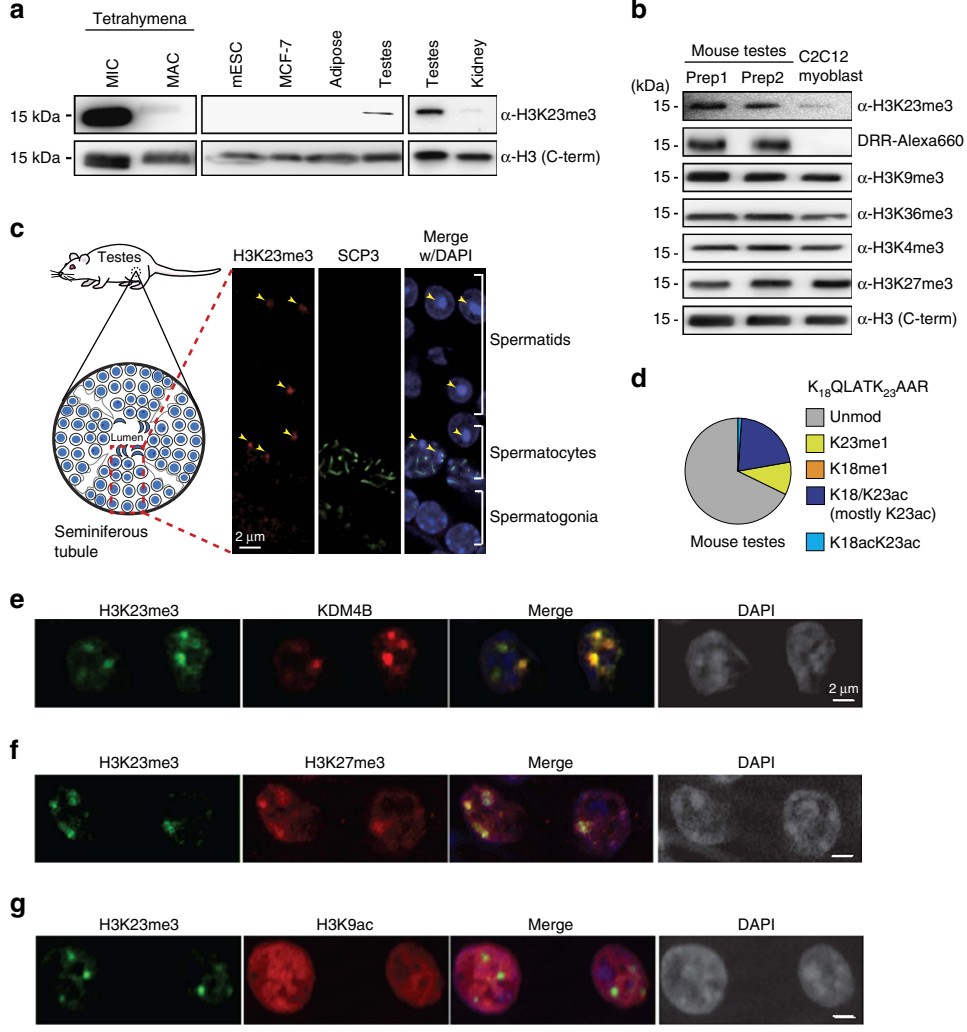

**Figure 5 | H3K23me3 is enriched in spermatocytes.** (**a**) Specific expression of H3K23me3 in different mouse tissues and mammalian cell lines. MIC (micronuclei) and MAC (macronuclei) histones from *Tetrahymena*, histones extracted from mouse tissues (testes, adipose and kidney) and mammalian cell lines (mESC, C2C12 and MCF-7) are probed with histone antibodies in a western blotting format. (**b**) Germline-enriched H3K23me3 detected by H3K23me3 antibody and DRR reader. Histones extracted from mouse testes tissues and C2C12 mouse myoblast cell line are probed with histone antibodies or engineered reader domain in a western or far western blotting format. (**c**) Immunofluorescence identifies H3K23me3 enrichment in meiotic cells. Mouse testes are probed with H3K23me3 (red) and SCP-3 (green) antibodies, and overlapped with 4,6-diamidino-2-phenylindole (DAPI) stain (blue; scale bar, 2 μm). Yellow arrows highlight H3K23me3 foci. (**d**) MS quantification of modifications on H3K23-containing peptide. (**e**) KDM4B (red) shows co-localization with H3K23me3 (green) by indirect immunofluorescence in rat testes (scale bar, 2 μm). (**f,g**) H3K23me3 (green) is localized at heterochromatin rather than active chromatin as suggested by co-localization with H3K27me3 (red in **f**) but not H3K9ac (red in **g**; scale bar, 2 μm).

somewhat more compact sub-nuclear distribution than in spermatocytes; however, H3K23me3 was nearly undetectable in mature sperms (Supplementary Fig. 5d). Meanwhile, H3K23me3 signal was much weaker in spermatogonia that have not entered meiosis (Fig. 5c and Supplementary Fig. 5d). The specificity of the H3K23me3 antibody was validated by peptide competition (Supplementary Fig. 5e). To further validate the higher methylation levels of H3K23 in mouse testes as detected by antibody and the recombinant reader probe, we also quantified different modification levels on H3K23 by mass spectrometry (MS; Supplementary Fig. 6). Indeed, we observed relatively higher global levels of H3K23 methylation ($\sim$10% using the quantified levels of monomethylation) in the testes (Fig. 5d) compared with levels ($<$1%) observed in other mammalian tissues or cell lines[29,34–36] (Levels of di- and trimethylation at H3K23 are difficult to quantify by MS due to the low global abundance in mammals). This observation provides further evidence on H3K23me3 being a conserved histone PTM associated with meiosis in mammalian systems.

Having established the physiological link of H3K23me3 with meiosis in mammalian spermatogenesis, we next investigated the *in vivo* connection between H3K23me3 and its putative reader, KDM4. As KDM4B harbours the unique specificity to read only the H3K23me3 modification (Fig. 4 and Supplementary Fig. 4d), we probed rat testes tissue to determine the extent of overlapping nuclear localization between KDM4B full-length protein and H3K23me3. A high degree of overlap was found between KDM4B and H3K23me3 staining in spermatocytes (Fig. 5e). The temporal and sub-nuclear overlap of both H3K23me3 and KDM4B in the same cell population suggests a relevant connection during meiosis. Another heterochromatic histone PTM, H3K27me3, was more broadly distributed than H3K23me3 across nuclei of germline cells; however, we found that in meiotic cells, a subset of H3K27me3 staining was highly correlated with the punctate distribution of H3K23me3 (Fig. 5f). In contrast, the euchromatic histone PTM H3K9ac is more diffused inside the spermatocyte nucleus and does not overlap with H3K23me3 (Fig. 5g). These immunofluorescence results map H3K23me3 to heterochromatin, but not euchromatin. Altogether, our results suggest that H3K23me3 helps target KDM4B within heterochromatin of mammalian meiotic spermatocytes.

**H3K23me3-binding stimulates H3K36 demethylation by KDM4B.** The co-localization of H3K23me3 and KDM4B *in vivo*, and the specific ability of KDM4B to bind H3K23me3 *in vitro*, prompted us to investigate the functional role of H3K23me3 binding by KDM4 proteins. KDM4A–C proteins are conserved histone lysine demethylases that have been linked to removal of H3K9me and H3K36me[5,6,8,13,37,38]. More recently, the catalytic domains of KDM4A–C were shown to demethylate H3K27 *in vitro* as well, although at a much slower rate[39]. Having determined binding specificity of the C-terminal DTD in KDM4B for H3K23me3 (Fig. 3), we reasoned that this interaction might enhance the activity of full-length KDM4B against certain histone methylations that co-occur with H3K23me3.

To investigate the functional consequences of KDM4 proteins binding to H3K23me3, we built several structural models that incorporate both the catalytic and the DTD. One structural model included KDM4A and dually modified peptide H3K4me3K9me3 (Supplementary Fig. 6a), and demonstrated the simultaneous engagement of H3K4me3 and H3K9me3. However, when the C-terminal reader domain binds H3K23me3, this binding mode requires the opposite orientation (head to toe) between histone H3 and KDM4 protein, suggesting that H3K27 and H3K36, but not H3K9, are more likely to be the demethylase target when

H3K23me3 is bound to the DTD (Fig. 6a). Our structural analysis shows no molecular interaction between KDM4B DTD and the H3K27 or H3K36 residue (Fig. 4), suggesting H3K23me3 binding should not directly impede H3K27 or H3K36 demethylation. We then built a structural model of KDM4B and H3K23me3K27me3 (Supplementary Fig. 6b), which provided evidence that either H3K27 or H3K36 is accessible for demethylation by the KDM4B catalytic domain.

To experimentally test the ability of KDM4B to demethylate H3K27me3 or H3K36me3 on H3K23me3 binding, we compared demethylation activity towards H3K27me3 or H3K36me3 by full-length KDM4B in the presence of H3K23me3 on the same histone peptide (Fig. 6a). Using MS/MS, we did not detect significant demethylation of H3K27me3 peptide by KDM4B, even after prolonged incubation (Supplementary Fig. 6c), consistent with previous reports of H3K27me3 being a poor substrate for the KDM4 family[5,6,8,13,37]. The lack of measurable demethylation on the dually modified H3K23me3K27me3 peptide (Fig. 6a) indicated that H3K23me3 binding by the reader domain did not stimulate the negligible H3K27me3 demethylation. In stark contrast, we observed consistent H3K36me3 demethylation by KDM4B (Fig. 6a and Supplementary Fig. 6c). Using a MS-based demethylation assay, we corroborated the findings: the co-existence of H3K23me3 and H3K36me3 on the same peptide yielded faster rates of H3K36me2 and H3K36me1 product formation, and a corresponding faster depletion of substrate (H3K36me3) compared with peptide with only H3K36me3 present (Fig. 6b). The MS analysis also confirms that increased activity of KDM4B in the presence of H3K23me3 is indeed due to demethylation of H3K36me3 and not the result of altered demethylation at H3K23me3. KDM4B preferentially demethylate H3K36me3 in the context of dually modified peptide when incubated with a 1:1 mixed pool of H3K23me3K36me3 and H3K36me3 peptide substrates (Fig. 6c). Consistent with the ability of both KDM4A and KDM4B but not KDM4C to bind H3K23me3 through its reader domain (Fig. 1), demethylation of H3K36 was enhanced for KDM4A and KDM4B but not KDM4C (Supplementary Fig. 7d,e). The fact that the H3K23me3K36me3 peptide displayed an accelerated rate of demethylation relative to the K36me3 peptide in the substrate competition experiment (Fig. 6c) supports a *cis* mechanism for demethylation by KDM4A and KDM4B.

Based on the observation that full-length KDM4B displays enhanced demethylase activity at H3K36me3 in the context of doubly modified H3K23me3K36me3, we determined the sub-nuclear distribution of KDM4B and H3K36me3 in rat primary spermatocytes (Fig. 6d). Consistent with the *in vitro* kinetic results, H3K23me3-specific KDM4B and H3K36me3 staining did not overlap and supports the efficient demethylation of H3K36me3 at these loci. Together, the results suggest that the interaction between H3K23me3 and KDM4B-DTD endows the demethylase with enhanced catalytic efficiency for removing H3K36me3 when these chromatin marks exist on the same histone tail.

## Discussion

In this study, we described the molecular basis and functional relevance of H3K23me3 recognition by the KDM4A and KDM4B DTDs. Biochemical analysis revealed the intricate molecular interactions that govern PTM specificity and sequence specificity of KDM4-DTDs for H3K23me3. In particular, we demonstrated the significance of side-chain interactions with residues surrounding the trimethyl lysine in mediating KDM4B-DTD's binding preference of H3K23me3 over H3K4me3. This study suggests a role for KDM4B in heterochromatin maintenance during meiosis,

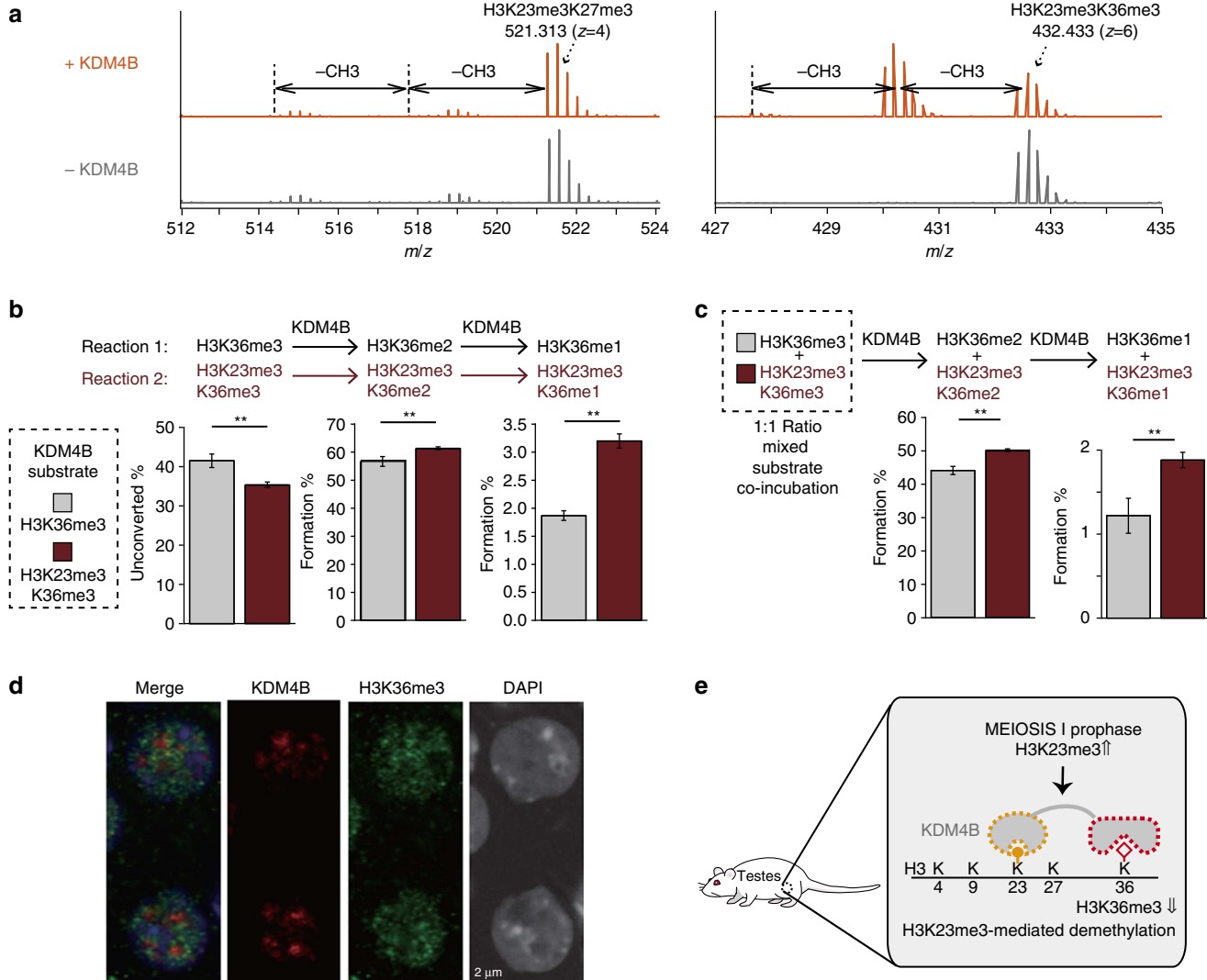

**Figure 6 | H3K23me3 binding stimulates H3K36 demethylation by KDM4B.** (**a**) MS analysis of demethylation reaction by KDM4B full-length protein. H3(17–34)K23me3K27me3 or H3(20–42)K23me3K36me3 peptides were used as substrates. Negative controls without KDM4B enzyme were shown side-by-side. The shown spectrums of only one charge state at one time point (H3K27me3 peptides: +4 charge state, 120 min; H3K36me3 peptides: +6 charge state, 30 min) display representative trends of all three charge states at three time points for each peptide. (**b**) Co-existing H3K23me3 increases H3K36me3 conversion to H3K36me1/2 by KDM4B. Product formation was measured by MS. Errors represent s.d. from three charge states of the same peptide. Student's *t*-test was performed (**P < 0.01). (**c**) KDM4B preferentially demethylates dually modified peptide in 1:1 (5 µM:5 µM) ratio mixture of H3K36me3 and H3K23me3K36me3 substrate. Site and level of demethylation was quantified by MS. Errors represent s.d. from three charge states of the same peptide. Student's *t*-test was performed (**P < 0.01). (**d**) KDM4B (red) shows distinct localization with H3K36me3 (green) by indirect immunofluorescence in rat primary spermatocytes (scale bar, 2 µm). (**e**) Proposed model of H3K23me3-mediated H3K36me3 demethylation by KDM4B during meiosis.

mediated by H3K23me3-dependent binding and demethylation of H3K36me3 (Fig. 6e). Lastly, our results support a conserved role of H3K23me3 during germ cell development.

KDM4 demethylases are conserved epigenetic regulators of histone methylation that are required for normal development[5–12]. The C-terminal DTDs in the multiple KDM4 homologues are only present in vertebrates (Supplementary Fig. 8), suggesting functional diversification as a result of genome duplication during evolution. The sophisticated intermolecular interactions present in KDM4-DTDs revealed by this study underlies the histone interactomes of human KDM4A–C DTDs and sheds light on distinct functions of KDM4 members[14–16]. The mild phenotype of individual KDM4 knockout mice[40,41] implicates some compensating functions among KDM4

members, which is consistent with overlapping histone PTM interactomes among KDM4A–C DTDs (Fig. 1). However, the H3K23me3-specific reader KDM4B-DTD displays poor H3K4me3 binding, whereas KDM4A-DTD and KDM4C-DTD have low-micromolar binding affinity with H3K4me3 (Fig. 1). This observation is in line with previous findings that KDM4C but not KDM4B overlaps with H3K4me3 genomic distribution and the fact that targeting of KDM4C is dependent on its DTD[16,40]. Altogether, the data suggest a model in which chromatin association of KDM4 members is mediated by both their divergent and common reader binding targets, to ensure correct epigenetic programming during development.

Germ cell chromatin is subject to epigenetic regulation[42,43]. Here we propose that H3K23me3 and H3K36me3 might be

involved in mammalian germ cell development, especially during the mitosis-to-meiosis transition, through cross-talk between the KDM4 reader and catalytic domains. We revealed H3K23me3 enrichment in mouse and rat testes, specifically in primary spermatocytes undergoing meiosis (Fig. 5). This provides further evidence about H3K23me3 being a conserved histone modification associated with meiotic chromatin, as previously identified in *Tetrahymena* and *C. elegans*[32]. The enrichment of H3K23me3 is also consistent with reduced immunostaining of H3K23ac during preleptotene to pachytene stage in mouse testes tissues[44]. The function and mechanism of this unique association with meiotic chromatin still needs further investigation. It remains to be experimentally tested whether H3K23me3 plays similar roles in meiotic DNA damage in both mammalian systems and *Tetrahymena*[32].

Previously, the KDM4 family and H3K36me3 were independently implicated in germ cell development[5,41,45]; however, the underlying relationship was not clear. Based on our detailed structural and biochemical analysis (Figs 2–4), we would postulate that the reader domain of KDM4B has evolved to discriminate for exclusive binding to H3K23me3 with possible regulation by PTMs on H3T22 and H3R26. The H3K23me3 binding by KDM4A and KDM4B is likely to be conserved across species, given the conservation of the H3K23me3-discriminating residue (N931 in human KDM4A and N951 in human KDM4B) as shown in Supplementary Fig. 8. The functional analyses reported here demonstrate that the distinct mode of H3K23me3 binding stimulates the demethyalse activity of KDM4B towards H3K36me3. These results are in accord with such 'cross-talk' between a histone methylation 'reader' and 'eraser', leading to *cis*-histone demethylation reported for PHF8, KDM7A[46,47], KDM4A and KDM4C[28,48]. In the case of PHF8, biochemical and structural characterization indicates H3K4me3 strongly promotes H3K9 demethylation[46]. Although our structural model and the preliminary competition kinetic assay suggests a *cis*-histone cross-talk mechanism between H3K23me3 binding and H3K27/K36 demethylation (Fig. 6c and Supplementary Fig. 7a,b), more detailed kinetic analysis will be needed to fully investigate *cis*-histone and *trans*-histone cross-talk between H3K23me3 and KDM4 demethylation sites (H3K9me, H3K36me and H3K27me), in particular using nucleosome substrates. Interestingly, another KDM4 family member, KDM4D, which lacks the C-terminal reader domain and can only demethylate H3K9, has also been implicated in spermatogenesis[41]. This suggests other histone PTMs (such as H3K9 methylation) might also be involved in germ cell development.

Interestingly, we found H3K23me3 also overlaps significantly with H3K27me3 by immunofluorescence in rat spermatocytes (Fig. 5f). This finding is consistent with previous MS results showing H3K23me3 physically co-exists on a subset of H3K27me3 modified histones in *Tetrahymena* and *C. elegans*[32,33], suggesting that similar positive 'cross-talk' between H3K23me3/H3K27me3 may occur in mammals as well. Furthermore, KDM4B binding to H3K23me3 was not disrupted in the presence of H3K27me3 (Supplementary Fig. 1b). Thus, the combined heterochromatic H3K23me3/H3K27me3 signature may be used to recruit KDM4B to remove H3K36me3 at heterochromatin during the initial stages of meiosis. Indeed, H3K27me3 and H3K36me3 are located in mutually exclusive regions of germline chromatin in *C. elegans*. Whether H3K23me3-dependent downregulation of H3K36me3 serves to reduce gene expression or increase meiotic DNA damage repair at heterochromatin needs further investigation. Collectively, these observations suggest enrichment of H3K23me3 during meiosis is conserved across multiple species and our molecular characterization provides a mechanistic basis for the future investigation of the relationship between KDM4B and H3K23me3 in the germline.

## Methods

**Plasmid construction.** KDM4 protein sequences used in this study correspond to Uniprot ID: O75164 (human KDM4A, isoform 1), O94953 (human KDM4B, isoform1) and Q9H3R0 (human KDM4C, isoform1). Plasmid encoding KDM4A DTD was kindly provided by R-M Xu (Chinese Academy of Sciences). Complementary DNA encoding human KDM4B-DTD was synthesized (Integrated DNA Technologies). Plasmid containing cDNA of full-length KDM4C (clone ID: HsCD00341594) was obtained from Harvard Plasmid Repository. DNA encoding DTDs (KDM4A 895–1011aa, KDM4B 917–1031aa and KDM4C 877–991aa) were cloned into pFN29 (Promega) with N-terminal (His)$_6$-HaloTag or pET28b with N-terminal (His)$_6$ Tag using In-Fusion HD cloning kit (Clontech). Plasmids encoding *Xenopus laevis* core histones were a generous gift of K. Luger (University of Colorado-Boulder). Site-directed mutagenesis was performed with Q5 Site-Directed Mutagenesis Kit (New England Biolabs).

**Peptide binding and MARCC.** *Protein purification.* HaloTag-tagged proteins were expressed in BL21 (DE3) cells and induced by 0.5 mM isopropyl-β-D-thiogalactoside at 20 °C–25 °C for overnight. The cell pellets were resuspended in 30 mM HEPES, 500 mM NaCl pH 7.4, with 1 mM dithiotheritol (DTT), 1 mM phenylmethylsulfonyl fluoride, 10 μg ml$^{-1}$ leupeptin, 10 μg ml$^{-1}$ aprotinin and 1 mg ml$^{-1}$ lysozyme. After agitation at 4 °C for 30 min, the cell resuspension was further lysed by three mild sonication cycles (20% amplitude, 5 s on, 5 s off, 1 min total per sonication cycle) with 1/4″ probe (Thermo Fisher). Clarified lysate supernatant was incubated with Ni-NTA resin (GE Life Sciences) in batches for 2 h, 4 °C. Resin-bound proteins were washed three times with 30 mM HEPES, 150 mM NaCl, 20 mM imidazole pH 7.4, followed by batch elution with 30 mM HEPES, 150 mM NaCl, 300 mM imidazole pH 7.4. Eluents were dialysed into 30 mM HEPES, 150 mM NaCl pH 7.4, supplemented with 3 mM DTT and up to 10% (v/v) glycerol. After being concentrated in 10 kDa molecular weight cutoff Amicon (Millipore), protein was quantified by Bradford assay and analysed on 12% SDS–PAGE followed by Coomassie staining (Supplementary Fig. 1a).

*Peptide microarray.* Peptide microarray was synthesized in-house[29,49]. Peptide arrays were handled with care and protected from light. All solutions were filtered before putting on the array. Assays were performed with a modified two-chamber simplex gasket (Intuitive Bioscience). After tightening the gasket onto the slide surface, the peptide array was blocked with blocking solution (1 × PBS, 0.05% Tween-20 pH 7.4, 1% BSA) at 4 °C overnight, to reduce nonspecific binding. (1) Reader binding: 10–50 nM purified recombinant HT-KDM4 DTDs were first reacted with excessive HaloTag ligand-biotin (Promega) in blocking solution on ice for 15 min and incubated on peptide array under mild rocking at 4 °C, 1 h. The slide was washed with PBS with 0.05% Tween (PBST) three times and incubated with 1:2,000 streptavidin-conjugated Alexa-Fluor647 (Invitrogen) in blocking solution at room temperature for 1 h. (2) Antibody binding: after blocking, 1:2,500–1:100 dilution H3K23me3 antibody in blocking solution were incubated with peptide microarray under mild rocking at 4 °C, 1 h. The slide was washed with PBST three times and incubated with 1:1,000 anti-rabbit IgG Alexa 647 (Cell Signaling) in blocking solution at room temperature for 1 h. For both assays, after three washes in PBST and a final wash in distilled water, the slide was dried by centrifugation and imaged at dual wavelengths of 532 and 635 nm on Axon GenePix 4000B (Molecular Devices). The laser power was set to 100%, with automatic gain adjustment (0.05% saturation tolerance) for dual photomultipliers. Image was obtained at 5 μm pixel resolution. Features in each block were defined by manual adjustment of 13 × 13 grid (feature diameter, 280 μm; column spacing and row spacing, 320 μm) to cover every spot. Signal intensities were quantified by GenePix Pro 6.1 software (Molecular Devices). For each spot (feature), the mean intensities for 635 nm wavelength were used for subsequent analysis. For each peptide species, an average was calculated from three replicate spots. The averaged intensities were normalized to the range between 0 and 1 by the lowest and highest values of selected peptides on each library. For comparison between peptides with combinatorial modifications, the intensities were normalized to the peptide with single modification as log$_2$ ratio. The signal at 532 nm wavelength was used to identify misprinting events.

*Fluorescence polarization analysis.* Peptide sequences are given in Supplementary Table 1. Fluorescein-labelled histone peptides (constant final concentration at 3 nM) were mixed with titrations of purified KDM4 DTD proteins (0.1–100 μM) in buffer (30 mM HEPES, 150 mM NaCl pH 7.4, 0.01% v/v Triton X-100). A no-protein control with only peptide probe was included. Polarization at each concentration was measured as triplicates in 384-well polystyrene black microplates (Thermo Fisher Scientific 262260) by Biotek Synergy H4 multimode plate reader (light source: xenon flash, offset from top: 7 mm, sensitivity: 60%, excitation: 485/20 nm, emission: 528/20 nm, both parallel and perpendicular, normal read speed). Fraction bound (Fb) at each concentration was calculated based on the corresponding polarization values (P): Fb$_c$ = ($P_c$ − $P_{min}$)/($P_{max}$ − $P_{min}$), where $P_{min}$ is the polarization value of no-protein control and $P_{max}$ is the polarization value of the saturation value for that peptide. Dissociation constants

$(K_d)$ were derived by KaleidaGraph, Synergy Software (version 4.1.3) using the following equation: $Fb = [protein]/([protein] + K_d)$.

*Matrix-assisted reader chromatin capture.* MCF-7 cells (ATCC) were cultured in DMEM medium supplemented with 10% fetal bovine serum. For each nucleosome isolation, MCF-7 cells at $\approx 90\%$ confluency from two 10 cm plates ($\approx 2 \times 10^7$ cells in total) were collected and washed for three times in ice-cold Buffer M (10 mM HEPES, 10 mM KCl, 1.5 mM $MgCl_2$, 340 mM sucrose pH 7.9, 10% glycerol, v/v), supplemented with 1 µg ml$^{-1}$ trichostatin A, 1 mM DTT, 0.5 mM phenylmethylsulfonyl fluoride, 10 mM β-glycerophosphate, 1 mM leupeptin and 1 mM aprotinin. At the last wash, the cell resuspension was lysed with 0.1% Triton X-100 on ice for 10 min. After lysis, nuclei pellets were resuspended in Buffer M and centrifuged at 1,300 g for 12 min through chilled sucrose cushion buffer (10 mM HEPES pH 7.9, 30% sucrose, w/v, 1.5 mM $MgCl_2$), to further purify the nuclei pellets. After three washes with Buffer M, nuclei was diluted to 1.2 to 1.6 mg ml$^{-1}$ DNA concentration and digested with 2,000 gel units of micrococcal nuclease (New England Biolabs) at 37 °C for 12 min with constant mixing in the presence of 1 mM final concentration of $CaCl_2$. Before assay, the amount of enzyme and digestion time was optimized to obtain above 90% purity of mononucleosomes. MNase activity was stopped with 10 mM EDTA and spun down. Soluble chromatin from the supernatant (S1) was collected. Less soluble chromatin (S2) was recovered from the nuclei pellets resuspended in 5 mM HEPES, 0.2 mM EDTA at 4 °C overnight. Pooled chromatin extract (S1 plus S2) was concentrated to $\approx 10$ µM and dialysed into 30 mM HEPES, 150 mM NaCl pH 7.4, 10% (v/v) glycerol. The matrix-assisted reader chromatin capture (MARCC) resins were prepared by incubation of saturating amounts (>1 nmol) of purified recombinant (HQ)5-HaloTag proteins with 20 µl HaloLink resin slurry (Promega) in MARCC buffer (30 mM HEPES, 150 mM NaCl pH 7.4, 0.01% NP-40, 10% glycerol) at 4 °C for 1 h. Excessive proteins were removed by three brief washes with MARCC buffer. Chromatin capture was achieved by incubating 100 pmol native mononucleosomes or reconstituted nucleosomes with MARCC resins under constant rotation at 4 °C overnight. Bound chromatin was further washed with MARCC buffer and eluted by Halo-TEV protease (Promega) cleavage in the presence of 20 µl 1 mM Tris-HCl pH 7.4 at room temperature for 2 h. Eluted chromatin was combined with another 20 µl resin-recovered chromatin and could be used for downstream analysis.

**Nucleosome reconstitution and MLA production.** MLAs were prepared as previously described[29]. Briefly, histone H3 (*X. laevis*) with K4C (or K23C) and C110A mutations was expressed in *E. coli* and purified by Superdex200 column followed by ion-exchange columns. The purified histone was reduced and the cysteine residue was alkylated with excess (2-bromoethyl) trimethylammonium bromide (Sigma-Aldrich) at elevated temperature. After the reaction was quenched with 2-mercaptoethanol, excess reagents and salts were removed using a PD-10 desalting device. Along with other core histones (H2A, H2B and H4) that are recombinantly prepared, MLA H3 was refolded into histone octamers. The purified octamer and 146 bp DNA fragments (shown to exhibit strong positioning to the histone octamers) were reconstituted into nucleosome core particle by a salt-gradient dialysis method[50].

**Protein crystallization.** *Protein purification.* For native protein production, the corresponding pET28a construct was transformed into the *E. coli* BL21 (DE3) and protein expression was induced in terrific broth media at 25 °C for 16 h after $OD_{600}$ reached $0.6 \sim 0.8$. The *E. coli* cells were harvested by centrifugation at 5,000 g for 15 min and re-suspended in buffer (20 mM HEPES, 150 mM NaCl, 10 mM imidazole pH 7.5). The cells were lysed by sonication on ice. Subsequently, N-His$_6$-KDM4A/B-DTD was purified via HisTrap HP column (GE Healthcare) with linear elution gradient of imidazole (10–500 mM). The His$_6$-tag of the proteins was cleaved by thrombin (Sigma) at 4 °C overnight in buffer (20 mM HEPES, 150 mM NaCl pH 7.5, 0.5 mM TCEP). Gel filtration (HiLoad 16/600 Superdex 200 pg, GE Healthcare) was performed in buffer (20 mM HEPES, 150 mM NaCl pH 7.5, 0.5 mM TCEP) to further purify cleaved proteins. The KDM4A-DTD and KDM4B-DTD were concentrated using Amicon Ultra-15 (3000 NMWL, Millipore) to 45 and 38 mg ml$^{-1}$, respectively, flash frozen in liquid nitrogen and stored at −80 °C.

*Crystallization.* Initial crystallization screens were performed using an in-house screen comprising IndexHT, SaltHT, Crystal Screen HT, PegRx (Hampton Research), MIDAS, Morpheus, JCSG$^+$ (Molecule Dimensions), Wizard screens 1–4 (Rigaku) and MCSG screens 1–4 (Microlytic) by the sitting drop method using a Mosquito dispenser (TTP labTech Hertfordshire, UK). The apo KDM4A crystals were grown by mixing 0.2 µl of protein sample solution (20–45 mg ml$^{-1}$, 20 mM HEPES, 150 mM NaCl pH 7.5, 0.5 mM TCEP) with 0.2 µl of reservoir solution 1 (100 mM sodium acetate pH 4.6 and 1.0 M ammonium tartrate) or reservoir solution 2 (1% (w/v) PEG-3350, 100 mM Bis-Tris pH 5.5 and 1.0 M ammonium sulfate) at 20 °C using the sitting drop method. For co-crystallization, purified KDM4A-DTD was mixed with the H3K23Me3 peptide (Peptide 2.0, sequence see Supplementary Table 1) at an $\sim$1:1.5 molar ratio and the mixture was incubated on ice for 1 h before used for crystallization at 20 °C. The complex crystal was grown in 100 mM CAPS and sodium hydroxide pH 10.5, 200 mM lithium sulfate, 1.2 M $NaH_2PO_4$ and 0.8 M $K_2HPO_4$. The apo KDM4B-DTD crystals were grown by mixing 0.2 µl of protein sample solution (15–38 mg ml$^{-1}$, 20 mM HEPES pH 7.5, 150 mM NaCl, 0.5 mM TCEP) with 0.2 µl of reservoir solution (60 mM $MgCl_2$ and $CaCl_2$, 100 mM Tris and Bicine pH 8.5, 10% (v/v) ethylene glycol and 20% (w/v) PEG 8,000) at 15 °C using the sitting drop method. All crystals mentioned above were then flash frozen in liquid nitrogen with additional 15–20% glycerol for data collection. Co-crystallization of KDM4B-DTD with H3K23me3 was also attempted in similar initial screens, but complex crystals were not obtained.

**Structure determination and homology modeling.** *Data collection and structure refinement.* X-ray diffraction data were collected at the Life Sciences Collaborative Access Team (LS-CAT) with an X-ray wavelength of 0.98 Å at the Advanced Photon Source (Argonne National Laboratory). Data sets were indexed and scaled using HKL[51] or XDS[52]. For phasing experiments on KDM4A-DTD and KDM4B-DTD, molecular replacement was used using 2QQR[23] as starting model and then *phenix.autobuild* was used for automatic model building[53]. All the structures were completed with alternating rounds of manual model building with COOT[54] and refinement with *phenix.refine*[53]. Structure quality was validated by Molprobity[55]. Structures were visually interpreted using a stereographic collaborative commodity 3D TV arrangement[56].

The H3K23me3-bound structure of KDM4A-DTD contains six subunits in the asymmetric unit: two subunits (Chain A, C) have clear electron density for H3Q19-K27, another two subunits (Chain B, D) have clear electron density for H3L20-R26 and the remaining two subunits (Chain E, F) are partially disordered, which lead to higher R-factors than expected (Table 1). This is consistent with the crystal-packing environment, where chain A–D can form a stable crystal lattice leaving a huge channel for chain E–F to fit in (Supplementary Fig. 2a). We used subunits A and C with better peptide coverage for analysis. Two apo structures of KDM4A-DTD at 1.99 and 2.15 Å (PDB ID: 5D6W, 5D6X; statistics summarized in Table 1) with the same space groups and better $R/R_{free}$ factors compared with previously reported KDM4A apo structures (PDB ID: 2GF7 and 2QQR)[23,30]. Analysis of the KDM4B structure reveals that the crystallographic asymmetric unit comprises two subunits with a limited interaction interface ($\sim$480 Å$^2$ by PISA)[31]. This result is consistent with size exclusion chromatography results (Supplementary Table 2), where the Stoke's radius of gyration of KDM4A–C DTDs indicates the protein exists in solution as a monomer. Our attempts to obtain a structure for KDM4C-DTD yielded crystals with limited diffraction even after optimization; therefore, we used the 1.56 Å resolution KDM4C-DTD structure solved previously (PDB ID: 2XDP)[57] as a template for modelling.

*Homology modelling:* based on the already known protein-peptide complex structures of KDM4A-DTD with either H3(1–7)K4me3 (PDB ID: 2GFA)[30] or H3(19–27)K23me3 (PDB: 5D6Y, in this paper), the two peptides were modelled into KDM4B-DTD (PDB ID: 4UC4) and KDM4C-DTD (PDB ID: 2XDP) separately. The HTD-2 domains of the KDM4A/B/C DTD were structurally aligned by PyMOL[58], to ensure a good alignment of aromatic cage and histone peptide binding residues: KDM4A-DTD (929–980 aa), KDM4B-DTD (949–1000 aa) and KDM4C-DTD (909–960 aa), respectively.

**Immunofluorescence microscopy.** *Mouse immunofluorescence staining.* Testes of 7-week-old male C56BL/6 mice were dissected and detunicated in PBS, fixed in 2% paraformaldehyde in PBS for 3 h at 4 °C, equilibrated into 30% sucrose solution overnight at 4 °C, embedded in O.C.T. (optimal cutting temperature) compound and then frozen. Seven-micrometre-thick sections were cut, adhered to slides, baked at 50 °C for 45 min and permeabilized in PBST (0.05% Triton X-100). Sections were blocked for 30 min in 10% normal goat serum and 0.3% BSA in PBST. Primary antibodies (rabbit anti-H3K23me3) were diluted 1:70 in PBS (with 0.1% BSA and 0.1% Tween-20) and incubated at 4 °C overnight. Secondary antibodies were goat anti-rabbit conjugated with Cy3 or Alexa-488 (Jackson Immunoresearch and Invitrogen) diluted at 1:200 in PBS (with 0.1% BSA and 0.1% Tween 20) and incubated for 2 h at room temperature. For peptide competition experiments, anti-H3K23me3 was incubated with 5 µg ml$^{-1}$ of indicated peptides for 1 h on ice and immune complexes were removed by centrifugation for 10 min at >16,000 g. Pre-absorbed primary antibodies were applied to the tissue samples for 1 h at room temperature, followed by secondary antibody incubation for 1 h at room temperature. Before imaging, DNA was counterstained with 4,6-diamidino-2-phenylindole. Confocal images were acquired with a Cascade QuantEM 512SC camera (Photometrics) attached to a Zeiss Axioimager microscope with Yokogawa spinning disk confocal scanner and Slidebook software (Intelligent Imaging Innovations). Maximal-intensity projections encompassing a single layer of nuclei were generated by the Slidebook software. Image processing was performed in Adobe Photoshop CS4.

*Rat immunofluorescence staining.* Testes of adult male Sprague–Dawley rats were dissected, fixed in 1% paraformaldehyde in PBS for 3 h at 4 °C, equilibrated into 15% sucrose solution overnight at 4 °C and transferred to 30% sucrose for 48 h. Testes were then embedded in Optimal Cutting Temperature Compound (Sakura) and 10 µm sections were cut, thaw-mounted on SuperFrost Plus slides and dried at room temperature. Sections were incubated in cold acetone/methanol (1:1) for 10 min and permeablized in PBS with 0.05% Triton X-100. Sections were blocked for 30 min in PBS containing 0.05% Triton X-100 and 3% BSA. Primary antibodies were diluted in PBS containing 0.05% Triton X-100 and 3% BSA as follows: rabbit α-H3K23me3 (1:80), rabbit α-KDM4B (1:500; ab191434; Abcam), rabbit α-H3K27me3 (1:500; 39155; Active Motif), mouse α-H3K36me3 (1: 1000;

C15200183; Diagenode) or rabbit IgG (1:250; 27295; Cell Signaling). The secondary antibodies used were Alexa Fluor 488-conjugated donkey anti-rabbit IgG (1:400; Jackson ImmunoResearch, 711-545-152) or Alexa Fluor 594-conjugated donkey anti-rabbit IgG (1:400; Jackson ImmunoResearch, 711-585-152) diluted at 1:400 in PBS containing 0.05% Triton X-100 and 3% BSA. Primary antibody incubations were kept overnight at 4 °C; secondary antibody incubations were kept for 1 h at room temperature. For dual staining of KDM4B/ H3K23me3 or K27me3/ H3K23me3, the H3K23me3 antibody was applied as the second primary antibody. With H3K23me3 staining alone, IgG was used in place of the first primary antibody. After incubation in the first primary antibody overnight (either α-H3K27me3, α-H3K27me3, α-KDM4B or α-IgG), sections were washed two times in PBS containing 0.05% Triton X-100 and incubated with Alexa Fluor 594-conjugated donkey anti-rabbit IgG (Jackson ImmunoResearch, 711-585-152) as the first secondary antibody for 1 h at room temperature. Following two washes with PBS containing 0.05% Triton X-100, sections were incubated with AffiniPure Fragment Goat Anti-Rabbit IgG (H + L) (1:70; Jackson ImmunoResearch, 111-007-003) for 1 h, to block the first secondary antibody. After two washes with PBS containing 0.05% Triton X-100, sections were incubated in the second primary antibody (α-H3K23me3) overnight at 4 °C. Sections were washed two times in PBS containing 0.05% Triton X-100 and incubated with Alexa Fluor 488-conjugated donkey anti-rabbit IgG (1:400; Jackson ImmunoResearch, 711-545-152). Sections were then washed two times with PBS containing 0.05% Triton X-100 and mounted in Fluoromount-G with 4,6-diamidino-2-phenylindole (00-4959-52; eBioscience). Digital images were captured using an inverted fluorescence microscope (AxioVision: Carl Zeiss) equipped with an ApoTome for optical sectioning and a camera (AxioCam MRM; Carl Zeiss). Image acquisition was performed with AxioVision software. Images were exported as tiff files and processed using Adobe Photoshop software CS6.

**Histone acid extraction from tissues.** Testes and other tissues were dissected in PBS from C57BL/6J male wild-type mice (3–6 months old) and homogenized in ice-cold hypotonic lysis buffer (10 mM Tris-HCl, 10 mM NaCl, 3 mM MgCl₂ pH 7.4) supplemented with proteinase inhibitors in a 1 ml dounce homogenizer. Homogenized tissue lysate was filtered through 100 µm nylon mesh cell strainer (Fisher Scientific) and pelleted (800 $g$, 10 min, 4 °C). Pelleted nuclei were washed with PBS twice and resuspended with 0.4 N H₂SO₄. After incubation at 4 °C for 4 h, the nuclear lysate was clarified by centrifugation (3,400 $g$, 5 min, 4 °C) and acid-soluble supernatant was precipitated with 100% trichloroacetic acid at 4 °C overnight. The precipitated proteins were rinsed with acetone + 0.1% HCl and the air-dried pellet was resuspended in water and quantified by Bradford assay (Bio-rad).

**Western blotting.** Purified recombinant MLA histones (100 pmol) or 1–2 µg acid extracted total histones were run on 12% or 15% SDS–PAGE and transferred to polyvinylidene difluoride membrane.

For antibody-based western blotting, the membrane was blocked with 5% milk in PBST and then incubated with primary rabbit polyclonal antibody (α-H3, 1:5,000; ab46765, Abcam; α-H3K4me3, 1:5,000; ab8580, Abcam; α-H3K9me3, 1:5,000; ab8898, Abcam; and α-H3K23me3, 1:100; from the Taverna lab[32]; α-H3K36me3, 1:1,000; ab9050, Abcam) or primary mouse monoclonal antibody (α-H3K27me3, 1:1,000; catalogue number 61017, Active Motif) at 4 °C, 4 h to overnight. Secondary antibody (horseradish peroxidase-linked goat-anti-rabbit or mouse, catalogue number 7074P2 or 7076S, Cell Signaling Technology) was incubated at room temperature for 1 h. Membrane was detected using ECL HRP SuperSignal West Dura Extended Duration Substrate kit (Thermo Scientific).

For reader-based far western, the membrane was blocked with 5% BSA in HBST (30 mM HEPES, 150 mM NaCl pH 7.4, 0.05% Tween 20) at room temperature for 1 h and then incubated with 100–200 nM HT-KDM4 DTD conjugated with HaloTag-Alexa 660 (Promega) at 4 °C overnight. All images were captured by ImageQuant LAS 4000 (GE Healthcare Life Sciences) and processed by ImageJ[59].

**Nano-liquid chromatography and electrospray ionization MS/MS.** Quantitative MS for histone PTMs was performed as previously described[34]. Histone extracts were derivatized using propionic anhydride for unmodified and monomethylated lysines, and phenylisocyanate for newly generated N termini on tryptic peptides as described[60]. Briefly, 5 µg of histone extract in 9 µl of water was buffered using 1 µl of 1 M triethylammonium bicarbonate. One microlitre of 1:100 propionic anhydride in H₂O was then added to each histone extract and incubated at room temperature for 2 min before a 20 min quench with 1 µl of 80 mM hydroxylamine. Histone extracts were trypsinized at 1:50 for 5 h at 37 °C, followed by addition of 1 µl 20 mM NaOH. Newly generated N termini were labelled by incubation with 3 µl of 2% phenylisocyanate/acetonitrile for 1 h at 37 °C and desalted using in-house prepared C18 stage tips. Approximately 0.5 µg of derivatized histone extract was injected onto a Dionex Ultimate3000 nanoflow HPLC with a Waters NanoEase C18 column (100 µm × 15 cm, 3 µm) coupled to a Thermo Fisher Q-Exactive MS at 700 nl min⁻¹. Mobile phase consisted of water + 0.1% formic acid (A) and acetonitrile + 0.1% formic acid (B). Histone peptides were resolved with a two-step gradient of 2–25% mobile phase B over

60 min followed by a gradient of 25–40% mobile phase B over 15 min. The MS was operated in data-dependent acquisition (DDA) mode with dynamic exclusion enabled (exclusion duration = 8 s), MS1 resolution = 70,000, MS1 automatic gain control target = $1 \times 10^6$, MS1 maximum fill time = 100 ms, MS2 resolution = 17,500, MS2 automatic gain control target = $2 \times 10^5$, MS2 maximum fill time = 500 ms and MS2 normalized collision energy = 30. For each cycle, one full MS1 scan range = 300–1,100 $m/z$, was followed by 10 MS2 scans using an isolation window size of 2.0 $m/z$. An inclusion list was employed to increase the detection efficiency of histone peptides of interest.

**Rodent experiments.** The rat tissues were obtained from adult wild-type Sprague–Dawley rats that were killed in accordance with existing Institutional Animal Care and Use Committee (IACUC)-approved protocols. The mouse tissues were obtained from wild-type C57BL/6 surplus male mice that were scheduled for termination under IACUC protocol and provided to our laboratory following killing. This procedure has been reviewed and approved by Johns Hopkins Animal Care and Use Committee and UW-Madison Animal Care and Use Committee (protocol number M02059) and was performed in accordance with the NIH Guide for the Care and Use of Laboratory Animals.

**MS-based analysis of peptide demethylation.** In a typical demethylase reaction, the assay mixture containing 50 mM Tris pH 7.5, 50 mM NaCl, 50 µM (NH₄)₂Fe(SO₄)₂, 0.5 mM ascorbic acid, 1 mM α-ketoglutarate and 0.2 µM full-length KDM4A–C (Active Motif) was mixed with 5–10 µM of trimethylated histone peptides (detailed sequence information in Supplementary Table 1). Each peptide concentration was quantified by measuring Abs₂₈₀ₙₘ for peptides containing either tryptophan or tyrosine residue. After incubation of assay mixture at 25 °C for 10, 30 and 60 min time point, 10 µl of aliquot was transferred to a clean tube containing 5 µl of 5% formic acid to quench the reaction. Each quenched sample was treated with standard StageTip protocol using C18 extraction disk (Model 2215, Empore, 3 M) to remove excess salts and detergents in the reaction mixture[61]. Each eluate from StageTip containing peptides was dried at a reduced pressure (vacuum centrifuge) and was re-suspended with 10 µl of 5% acetonitrile solution (0.1% formic acid). The 1 µl of the sample was injected at 0.7 µl min⁻¹ flow rate onto a Dionex Ultimate 3000 Nano LC system equipped with Atlantis dC18 NanoEase reverse-phase column (3 µm, 100 µm × 150 mm, Waters) and coupled to a hybrid quadrupole-orbitrap MS (Q-Exactive, Thermo Scientific). The peptide was eluted at 4–15 min retention time with a linear gradient of 2–35% acetonitrile (0.1% formic acid). All the solvents used for liquid chromatography MS were HPLC grade (Fisher). For each analysis, MS1 scan range of 300–1,100 $m/z$ was used and data-dependent MS/MS was obtained to identify the site of demethylation. Each peptide peak and its area integration was performed on the XCalibur Qual Browser.

**Data availability.** X-ray structures have been deposited in the Protein Data bank with PDB ID: 4UC4, 5D6Y, 5D6W and 5D6X.

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

## Acknowledgements

We thank all Denu Lab members for their support. We thank Gene Expression Center (University of Wisconsin Biotechnology Center) for technical assistance with peptide array printing and scanning, and LS-CAT beamline at the Advanced Photon Source (Argonne National Laboratory) for help in collecting the diffraction data. We thank Jue Zhang (University of Wisconsin at Madison) for collecting mouse tissues. We thank Khoa Tran (University of Wisconsin at Madison) for the mESC histones. We thank Peter W. Lewis, Rupa Sridharan and Xuehua Zhong for helpful discussion of the manuscript. This research was supported by NIH grant numbers 2R37GM059785-15/P250VA (J.M.D.) and R01GM106024 (S.D.T.), Protein Structure Initiative grants U01 GM098248 (G.N.P. Jr.) and American Heart Association Postdoctoral Fellowship 15POST25680060 (J.-H.L.).

## Author contributions

Z.S., F.W., J.-H.L., S.D.T., G.N.P. Jr. and J.M.D. conceived and designed the study. F.W., M.D.M. and G.N.P. Jr. obtained crystals and solved the structures. Z.S. carried out most of the molecular and biochemical experiments. F.W. and Z.S. performed structural modelling and analysis. J.-H.L. purified MLA histones and performed demethylase assays. K.E.S., R.P., E.V. and S.D.T. generated H3K23me3 antibody and performed immunofluorescence on rodent testes. K.A.K. performed histone PTM MS analysis. A.R. made KDM4B aromatic cage mutant. J.J.T. purified histones from Tetrahymena. M.D.B. and V.I.K. synthesized and purified peptides. Z.S. and K.A.K. extracted histones from mouse tissue and cell lines. Z.S., F.W., S.D.T. and J.M.D. drafted the manuscript. All authors discussed the results and commented the manuscript.

## Additional information

**Competing financial interests:** J.M.D. consults for Bio-Techne and FORGE Life Science. The remaining authors declare no competing financial interests.

