## [Peer Review File · Nature Communications]

Reviewer #1 (Remarks to the Author):

This is an interesting, clearly written paper reporting an apparent unique activity of KDM4B in recognizing the H3K23me3 epigenetic mark, which enhances the catalytic activity of the enzyme for its previously reported substrate H3K36me2/3. The authors go on to link this activity to meiosis and spermatogenesis in mammalian cells. The experiments seem to be solid, especially the structural biology and biochemistry - my area of expertise. (I cannot judge the mouse/spermatocyte data as rigorously).

The results are important and should be published. They demonstrate a novel activity within the KDM4A family with careful delineation of the differences within the double tudor domains of this family. They further provide some of the first evidence for a role for H3K23me3 in mammalian biology and link this to KDM4B with a plausible mechanism.

I have some questions and comments that may help with clarity.

1. The authors describe in multiple sections of the manuscript the 'molecular network' that defines or discriminates histone peptide recognition by the DTD domains of KDM4A-C. The term molecular network connotes interactions between molecules, such as protein-protein interaction networks. I think what they are discussing in these sections might be more accurately describes as the structural basis for peptide recognition/discrimination. (It is a network of atomic interactions among the side chains of DTD and H3 peptide, and only involves 2 'molecules', so was a confusing term for me).
2. The H3K23me3 mark is located in only a sub-region of the H3K27me3-marked heterochromatin according to the figure 5f. Could this be a unique functional region? The punctate foci of H3K27me3 suggest that only a small region of the genome is regulated by this mark.
3. On page 16 KDM4B is described as harboring 'the unique ability to read the H3K23me3 modification'. However KDM4A also recognizes this mark. The wording should be changed to something like 'Because KDM4B reads only the H3K23me3 mark....'

Reviewer #2 (Remarks to the Author):

In this manuscript by Zhangli Su et al, the authors identified the Double Tudor domains of KDM4 family demethylases as an H3K23me3 reader module through peptide binding and co-crystallography, and provided convincing evidence that the peptide substrate Km of KDM4B, in particular, can be enhanced by pre-existing H3K23me3 on the substrate, providing another

previously unknown case of modification crosstalk. In general, the findings are of high interest, as H3K23me3 is a rare histone methylation and the field knows very little about it. The data are well organized and presented and the manuscript is nicely written.

Specific comments

1. The authors need to show the specificity of the in-house made H3K23me3 antibody. As several pieces of the functional data were produced using the antibody, and the confidence of the data is dependent on the specificity of this antibody. The authors should use the peptide array developed by themselves to test the specificity and include the result as supplemental data.

2. To further strengthen the point of the crosstalk and to complement the Km study, it would be nice to determine whether the H3K23me3 binding mutant of KDM4B has impaired function when compared to wildtype? For example, the ability of co-localization of KDM4B and H3K23me3 in cell can be tested, and the demethylase activity either in vitro or in cell after transfection can also be tested.

Reviewer #3 (Remarks to the Author):

Su et al. identify and characterize the conserved double tudor domain (DTD) of KDM4 family members as a reader module for H3K23me3, a histone modification for which little is known. KDM4A, in addition to its known ability to bind H3K4me3 and H4K20me3, binds H3K23me3 while KDM4B exclusively binds H3K23me3 with high affinity. Structural analysis of KDM4A in complex with an H3K23me3 peptide revealed residues that are necessary for H3K23me3 interaction. By comparative analysis of KDM4A and KDM4B, the authors identified residues in KDM4B that provide exclusive binding preference for H3K23me3. To determine the functional role of KDM4B as an H3K23me3 reader, the authors identified mouse testes as a tissue with enriched levels of H3K23me3. KDM4B was determined to localize to regions of H3K23me3. Finally, binding of H3K23me3 by full-length KDM4B stimulated demethylation of H3K36me3 in vitro. The authors speculate that this mechanism is in place during meiosis to remove H3K36me3 and promote a heterochromatic state.

This manuscript is of interest, as little is known about H3K23me3, and the identification of the KDM4 family as readers of this mark provide potential mechanistic understanding of the role of this histone modification in chromatin organization. While the authors provide convincing evidence for the binding of H3K23me3 by KDM4 family members and comprehensive biochemical characterization of the intramolecular network that contributes to this interaction, several concerns should be addressed to support the major conclusions of the study.

Major concerns:

1. The authors use antibody capture followed by immunofluorescence to demonstrate antibody specificity for their in-house H3K23me3 antibody (Supplementary Figure 5b). However, only 3 additional histone marks (H3K27me3, H3K9me3, H3K4me3) were assayed to demonstrate specificity for H3K23me3. Given the conclusions drawn from the use of this antibody in Figure 5, more comprehensive specificity profiling of the H3K23me3 antibody needs to be done.
2. A major conclusion is that H3K23me3 is enriched in primary spermatocytes undergoing meiosis; however, only Figure 5c is presented in support of this conclusion, and the H3K23me3 antibody also detects this mark in spermatids. This conclusion would be strengthened if a quantitative analysis of H3K23me3 among the different cell types was provided.
3. While the current data does support that H3K23me3 stimulates demethylation of H3K36me3 in vitro, the experiment presented is not able to discriminate between cis and trans interactions of KDM4B with these two histone modifications. Additional experiments need to be performed to support the claim of cis regulatory function. In addition, demethylase assays should be done with a characterized mutant that disrupts H3K23me3 interaction.
4. Cellular co-localization of KDM4B with H3K23me3 would be strengthened by assaying a DTD mutant unable to bind this mark.

Minor concerns:

1. For the fluorescence polarization (FP) experiments, only H3unmod (1-14) is listed as being used in FP. However, as the main focus of the paper is related to H3K23me3, a H3unmod (17-32) should also be used as control in binding experiments.

2. For results provided on the histone peptide microarray, only the tri-methyl states for H3K23 are provided. Given the FP results with regard to di- and mono-methyl H3K23me₃, it would be useful to the reader if the signals generated from these modifications were also provided. In addition, a more comprehensive analysis of the microarray data should be presented.

3. In the methods section for the Mass Spectrometry-based demethylation assay, it indicates that this assay was performed with KDM4A-C; however, only the results from the KDM4B assay are presented. What were the results for KDM4A and KDM4C?

Minor typos:

1. In Figure 2k, the Figure legend indicates that H3(19-27) is shown, but only H3(20-26) is shown in the figure.

2. In Figure 5A, the text states that H3K23me₃ was profiled across adipose, testes, kidney, and liver. However, 2 lanes are labeled as testes and 0 lanes are labeled as liver.

3. Some of the phrasing on pg. 18 is unclear:

Line 3,4,5: the authors state "we compared demethylation activity toward H3K27me₃ or H3K36me₃ by full-length KDM4B in the presence or absence of H3K23me₃ on the same histone peptide (Figure 6b, Supplementary Fig. 6c)." However, Figure 6b shows that presence or absence of KDM4B with H3K23me₃ in both experiments, while Supplementary Fig. 6c shows the demethylation assay with the absence of H3K23me₃ in either the presence or absence of KDM4B. The sentence referring to these two different experiments needs to be clarified as it currently does not convey what the figures demonstrate.

Line 9-11: the authors state: "The lack of measurable demethylation on the dually modified H3K23me₃K27me₃ peptide (Fig. 6b) indicated that the negligible demethylation of H3K27me₃ was not be stimulated by H3K23me₃ binding." This sentence needs clarification.

Line 22: the authors state: "depletion of substrate (H3K23me₃)", but the substrate that is being depleted is H3K36me₃.

Reviewer #4 (Remarks to the Author):

This is a potentially interesting manuscript for Nature Communications. The biochemistry is the strength of the work. In regard to this I think it is essential to show the identified H3K23me3 tudor domain's alter regulate KDM4 KDM activity in the context of full length KDM/nucleosome interactions. Looking at the effect (or not) with respect to H3K9me3 activity (along with H3K36me3, the best characterised of the KDM4 activities) is essential. Specific experiments are outlined below. If this can be done (all possible) the next step is to demonstrate (via CHIP type analyses and appropriate materials) that the normal targeting mechanism is valid in cells (this could be in any organism from my perspective - yeast is OK). This would make a nice story.

Where I struggled with the 'story' was the connection of the biochemical work through to spermatogenesis/meiosis. As I read the paper the link to an H3K27/KDM4 tudor domain and KDM catalysis has not yet been made in vivo.

The abstract is a bit misleading here with respect to "in vivo co-localisation"; this implies a direct interaction that has not actually been demonstrated.

I don't think it is necessary to make a full connection to physiology, but I think it needs to be made clear that this is, as yet, an undemonstrated possibility.

I think aspects of the manuscript could be shortened, and it could be reorganised, separating biochemistry/structural biology, cellular interaction work and biological results. Some key primary experimental data, e.g. for the histone MS studies (which are non-trivial), is also missing and should be included.

Specific Points

Title: Possibly 'activity' or 'site of action' is better than 'functions'.

Abstract:

"...specific chromatin regions"?

Replace 'novel epigenetic mechanism' with 'a previously unidentified targeting mechanism'. I am unsure if the role in spermatogenesis (and meiosis) has been sufficiently validated to mention in abstract - no need to over-claim.

P5. Demethylase subfamily.

Perhaps mention there are other KDM4s with different selectivity (KDM4D).

The paragraph on 16 needs to be re-written - it presently reads as another abstract. This should set out the rationale of the work and key results.

The introduction could mention that the demethylation activity of some KDMs is known to be directed by methylation at non-demethylated sites, e.g. PHF8.

P7. Express genes, not proteins.

P7. Mention binding assay and in the micro array.

P7/8. I think it is important to support the fluorescence results with ITC or other established solution assays for binding (e.g. NMR).

P8. It is difficult to distinguish between cation- π and hydrophobic interaction - see recent reported work on PHD domains.

P8. Electron density maps must be shown for all the complexed structures.

P10/11/12. I do appreciate that this is (in one aspect) principally a biochemical study, but I do think it is important to validate the relevance of the interactions within (at least) the context of 'full length' KDM4 - nucleosome interactions. In this regard the statement on validation re biochemistry validating the interactions is misleading - it is biological validation that is important.

In this regard the results in Supp Fig 6a are crucial - I would separate these from the biochemical work and put in the main text. It is also essential to show the interaction is valid using full length KDM4 (OK to focus on one KDM4 and so more in depth assays rather than several KDM4s) and, appropriately, modified nucleosomes. The nucleosomes could be recombinant - selectively modified by enzymatic or chemical means, using wildtype and mutant KDM4s.

It is also important to test whether the potential new binding interactions influence the KDM activity of the KDM4s.

P14. The work on the biological role of H3K23me3 is interesting. More detail, including primary data (including good quality MS/MS spectrum |) is required for the MS analysis. Such studies are non-trivial, especially to do quantitatively.

P16. As noted above the potential unique ability (what is the evidence it is unique?) needs to be validated in more biologically relevant contexts. What is needed in the cellular analysis is not co-localisation but evidence for a direct interaction, using standard (but not necessarily easy) IP methods.

P17. The evidence for the co-occurrence of H3K23me3 and other markers needs to be defined - Fig.6a is a cartoon!

P17. The 'modelling' paragraph could be cut/moved to Discussion. Easiest just to report the experimental evidence.

P18. The results that H3K36me3 demethylation is enhanced by H3K23 methylation resulting from a reduction in Km is interesting.

It is also important to test for an effect on the other established KDM4 substrate, i.e. H3K9me3/2. Also important to demonstrate that the increased activity is due to the K3K23/KDM4 interaction by appropriate mutagenesis studies.

Discussion

I would focus on biochemical results first - as indicated above with discussing in the context of preliminary work that catalytic KDM activity is targeted by binding to non-catalytic domains.

Overall, although the results on H3K23 in meiosis / spermatogenesis are interesting, I am not sure they are sufficiently well connected to the biochemical results. If this connection is to be made, at least, good quality evidence for the targeting of KDM4 KDM activity by H3K27 methylation in cells needs to be presented.

Author response for

Title: Reader Domain Specificity Directs Lysine Demethylase-4 Family Function

Author: Zhangli Su, Fengbin Wang, Jin-Hee Lee, Kimberly E. Stephens, Romeo Papazyan, Ekaterina Voronina, Kimberly A. Krautkramer, Jeremy J. Thorpe, Melissa D. Boersma, Vyacheslav Kuznetsov, Mitchell D. Miller, Sean D. Taverna, George N. Phillips Jr., John M. Denu

Overall response:

We thank all the reviewers for their appreciation and constructive comments. We are very excited to see the positive comments from all 4 reviewers about our results as 'important and should be published' (Reviewer #1), 'of high interest' and 'nicely written' (Reviewer #2), 'of interest' (Reviewer #3), and 'potentially interesting manuscript for Nature Communications' (Reviewer #4). We are also very pleased to see their appreciation of the structural biology and biochemistry in our study as 'solid' (Reviewer #1), 'convincing' and 'comprehensive' (Reviewer #2), and 'the strength of the work' (Reviewer #4). Below we address each reviewer's comments/concerns in a point-by-point format. All edits have been included in the revised manuscript as well.

Underline denotes reviewers' comments

Red font or ***yellow shade*** denotes highlighted words

Reviewer #1 (Remarks to the Author):

This is **an interesting, clearly written paper** reporting an apparent unique activity of KDM4B in recognizing the H3K23me3 epigenetic mark, which enhances the catalytic activity of the enzyme for its previously reported substrate H3K36me2/3. The authors go on to link this activity to meiosis and spermatogenesis in mammalian cells. **The experiments seem to be solid, especially the structural biology and biochemistry - my area of expertise.** (I cannot judge the mouse/spermatocyte data as rigorously).

The results are important and should be published. They demonstrate a novel activity within the KDM4A family with careful delineation of the differences within the double tudor domains of this family. They further provide some of the first evidence for a role for H3K23me3 in mammalian biology and link this to KDM4B with a plausible mechanism.

I have some questions and comments that may help with clarity.

1. The authors describe in multiple sections of the manuscript the '**molecular**

network' that defines or discriminates histone peptide recognition by the DTD domains of KDM4A-C. The term molecular network connotes interactions between molecules, such as protein-protein interaction networks. I think what they are discussing in these sections might be more accurately describes as the structural basis for peptide recognition/discrimination. (It is a network of atomic interactions among the side chains of DTD and H3 peptide, and only involves 2 'molecules', so was a confusing term for me).

Response: We thank this reviewer for his/her comment on the wording of 'molecular network'. As Reviewer #1 suggested, we did mean the network of atomic interactions (both side-chain and main-chain) between two molecules, i.e. the reader domain and the H3 peptide. To avoid any confusion to do with protein-protein interaction network, we have now followed Reviewer #1's suggestion and avoided the use of 'molecular network' in the revised manuscript (3 occurrences in total). We have changed 'molecular network' into 'intermolecular interactions' as follows.

(1) Page 4, Highlights, bullet point 2:

'Structural and biochemical analysis identifies the **intermolecular interaction network** for specific H3K23me3 recognition.'

changed into

'Structural and biochemical analysis identifies the **intermolecular interactions** for specific H3K23me3 recognition.'

(2) Page 12, Section 'KDM4B-DTD is an H3K23me3 specific reader domain':

'..., we investigated whether KDM4B-DTD engages H3K23me3 using a **molecular network** similar to KDM4A-DTD.'

changed into

'..., we investigated whether KDM4B-DTD engages H3K23me3 using **intermolecular interactions** similar to KDM4A-DTD.'

(3) Page 20, Section 'Distinct and overlapping histone PTM interactomes among KDM4 family':

'The sophisticated **molecular network** present in KDM4-DTDs revealed by this study...'

changed into

'The sophisticated **intermolecular interactions** present in KDM4-DTDs revealed by this study...'

2. The H3K23me3 mark is located in only a sub-region of the H3K27me3-marked heterochromatin according to the figure 5f. Could this be a unique functional region? The punctate foci of H3K27me3 suggest that only a small region of the genome is regulated by this mark.

Response: We thank the reviewer for pointing this out. The reviewer is right about H3K23me3 only located in sub-region of H3K27me3-marked heterochromatin. This is also consistent with what we and others have observed in *Tetrahymena* and *C. elegans* (Methylation of histone H3K23 blocks DNA damage in pericentric heterochromatin during meiosis, 2014 *Elife*, Figure 1, 4 and 7; see also Dynamic changes of histone H3 marks during *Caenorhabditis elegans* life cycle revealed by middle-down proteomics, 2016 *Proteomics*). To better emphasize such fact, we have edited the text accordingly:

Page 16:

'Another heterochromatic histone PTM, H3K27me3, was **more broadly** distributed **than H3K23me3** across nuclei of germline cells; however, we found that in meiotic cells, a subset of H3K27me3 staining was highly correlated with the punctate distribution of H3K23me3 (Fig. 5f).'

Page 22 (Discussion):

'Interestingly, we found H3K23me3 also overlaps significantly with H3K27me3 by immunofluorescence in rat spermatocytes (Fig. 5f). This finding is consistent with previous mass spectrometry results showing **H3K23me3 physically co-exists on a subset of H3K27me3 modified histones in *Tetrahymena* and *C. elegans***^{32,33}, suggesting that similar positive "crosstalk" between H3K23me3/H3K27me3 may occur in mammals as well.'

3. On page 16 KDM4B is described as harboring 'the unique ability to read the H3K23me3 modification'. However KDM4A also recognizes this mark. The wording should be changed to something like 'Because KDM4B reads only the H3K23me3 mark....'

Response: We thank Reviewer #1 for this suggestion. We intend to highlight two ideas with this sentence: (1) KDM4B reads only the H3K23me3 mark; (2) KDM4B specificity is different (unique) from KDM4A and KDM4C. Together these two points justify the reason why we chose KDM4B but not KDM4A or KDM4C to do the following immunofluorescence experiment. To clarify this sentence in compliance with Reviewer #1's suggestion, we have edited this sentence as follows.

Page 16, second paragraph:

'Because KDM4B harbors the unique ability to read the H3K23me3 modification, ...'

changed into

'Because KDM4B harbors the unique **specificity** to read **only** the H3K23me3 modification, ...'

Reviewer #2 (Remarks to the Author):

In this manuscript by Zhangli Su et al, the authors identified the Double Tudor domains of KDM4 family demethylases as an H3K23me3 reader module through peptide binding and co-crystallography, and provided convincing evidence that the peptide substrate Km of KDM4B, in particular, can be enhanced by pre-existing H3K23me3 on the substrate, providing another previously unknown case of modification crosstalk. In general, the findings are of high interest, as H3K23me3 is a rare histone methylation and the field knows very little about it. The data are well organized and presented and the manuscript is nicely written.

Specific comments

1. The authors need to show the specificity of the in-house made H3K23me3 antibody. As several pieces of the functional data were produced using the antibody, and the confidence of the data is dependent on the specificity of this antibody. The authors should use the peptide array developed by themselves to test the specificity and include the result as supplemental data.

Response: We thank Reviewer #2 for this suggestion. As the reviewer pointed out, we understand it is important to profile the specificity of our in-house made H3K23me3 antibody given its usage in several functional experiments. In our original manuscript where this antibody was first described (original Supplementary Fig. 5b, now Supplementary Fig. 5e), we had performed peptide competition (antibody pre-incubated with indicated peptides) to demonstrate the antibody specificity in immunofluorescence assays. We reason this is an important control experiment, since immunofluorescence is one of the key functional approaches we used with H3K23me3 antibody (Fig. c, e-g).

Furthermore, in the previous 2014 Elife paper (now compiled into For Review Only - Figure 1), we had performed ELISA analysis on this same antibody against nine histone peptides (including H3K23me3, H3K23me2, H3K23me1, H3K27me3, H3K23me3K27me3, H3K4me3, H3K9me3 and H3unmod). We also performed peptide competition experiments with histone peptides containing common histone PTMs (including H3K4me3, H3K9me3 and H3K27me3) in ELISA format (FRO-Figure 1c). Overall, this previously published ELISA analysis also provides support that our in-house made antibody is strongly selective for H3K23me3 versus other well-studied sites of histone methylation.

For Review Only- Figure 1

Figure 2 (b-c) and Figure 2 -Supplementary Figure 1 in
Methylation of histone H3K23 blocks DNA damage in pericentric heterochromatin during meiosis.
Papazyan R, Voronina E, Chapman JR, et al. *Elife* (2014).

To further validate antibody specificity with a more systematic and high-throughput assay, we have now followed the reviewer's suggestion to profile this antibody with our peptide microarray that covers over 1000 unique histone peptide species (these new results are now included in Supplementary Fig. 5b-c). Among over 1000 unique peptide species on our peptide microarray, the H3K23me3 antibody displays a great specificity towards H3K23me3-containing peptides (Supplementary Fig. 5b-c). On the scale from zero (no binding) to one (highest binding), only the H3K23me3-containing peptides give relative signal intensities of more than 0.3 (Supplementary Fig. 5c), demonstrating great specificity for our H3K23me3 antibody. We observe negligible off-targets (relative intensity of less than 0.1) to other common Kme3 sites (including H3K4me3, H3K9me3, H3K27me3, H3K36me3, H3K56me3, H3K64me3, H3K79me3, H3K115me3 and H4K20me3). As expected from our ELISA data, minor cross-reactivity (relative intensity of 0.1 ~ 0.3) was observed for H3K23me2-containing peptides. Taken together, we have employed three different assays (ELISA, peptide microarray and peptide competition on immunofluorescence) to evaluate specificity of the H3K23me3 antibody used in this study. Uniformly, this data suggests the H3K23me3 antibody is a useful and unambiguous tool for gauging the presence and localization of H3K23me3 in chromatin. We have now edited the main text (shown below) and method section to include our new results.

Page 15 first paragraph:

“Using an H3K23me3-specific antibody produced in-house³² and profiled on peptide microarray (Supplementary Fig. 5b-c), ...”

2. To further strengthen the point of the crosstalk and to complement the Km study, it would be nice to determine whether the H3K23me3 binding mutant of KDM4B has impaired function when compared to wildtype? For example, (1) the ability of co-localization of KDM4B and H3K23me3 in cell can be tested, (2) and the demethylase activity either in vitro or in cell after transfection can also tested.

Response: We thank the reviewer for this suggestion. (1) The cellular co-localization of KDM4B and H3K23me3 (Fig. 5e) was performed on fixed rat testes tissue section. The suggested experiment could be done by introducing a point mutation into KDM4B in the wild-type mouse/rat, which will not be feasible in the given time frame. The alternative of using an H3K23me3-positive cell line is impossible at this point - we have assayed an array of mammalian culture cell lines (data not shown) for endogenous H3K23me3 expression and have not found one. We reason this is probably due to the specific expression pattern of H3K23me3 (enriched in meiotic and post-meiotic cells). Moreover, the enzymes catalyzing addition or removal of H3K23me3 have not been identified yet, so we can't manipulate H3K23me3 levels in an H3K23me3-negative cell line. Currently, very little is known about H3K23me3, which makes the suggested experiment very difficult. We believe the suggested experiment will be feasible in the future with more characterization of H3K23me3, but is beyond the scope of this current study.

To address the reviewer's concern (to strengthen the conclusion of co-localization between KDM4B and H3K23me3), we have performed the following new set of experiments: (1) First, we generated the aromatic cage mutant of KDM4B-DTD (Y993A), and demonstrated its dramatic decrease of H3K23me3-binding ability by FP ($K_d = 327 \mu\text{M}$, Supplementary Fig. 4d). (2) We showed wild-type KDM4B-DTD but not the Y993A mutant can recognize H3K23me3 MLA histone in a far western blot (Supplementary Fig. 4e). (3) We also showed only the wild-type KDM4B-DTD but not the Y993A mutant can pull-down H3K23me3-containing MLA nucleosomes (Fig. 4d). We believe that our new data using more physiologically relevant chromatin substrate (nucleosomes) in far western blots and nucleosome pull-down further supports our conclusion that KDM4B binds H3K23me3 via its reader domain.

(2) As for assaying demethylase activity with binding mutant, we have not been able to make mutant full-length KDM4 proteins. But to support our conclusion (H3K23me3 stimulates H3K36me3 demethylation by KDM4B), we have included a set of new experiments in which we incubated KDM4 enzymes with a 1:1 mixture of H3K36me3 and H3K23me3K36me3 substrates and monitored

demethylation conversion over time ('co-incubation assay' in Fig. 6c and Supplementary Fig. 7d-e). Together with Fig. 6b, the new results demonstrate that KDM4B (and KDM4A) demethylates H3K36me3 faster in the presence of H3K23me3; in other words, H3K23me3 stimulates demethylation of H3K36me3 for KDM4A and KDM4B, which bind H3K23me3, but not KDM4C, which does not bind H3K23me3. The results of this MS-based assay are now described in the text as below.

Page 19:

'KDM4B preferentially demethylate H3K36me3 in the context of dually modified peptide when incubated with a 1:1 mixed pool of H3K23me3K36me3 and H3K36me3 peptide substrates (Fig. 6c). Consistent with the ability of both KDM4A and KDM4B but not KDM4C to bind H3K23me3 through its reader domain (Fig 1), demethylation of H3K36 was enhanced for KDM4A and KDM4B but not KDM4C (Supplementary Fig. 7d-e). The fact that the H3K23me3K36me3 peptide displayed an accelerated rate of demethylation relative to the K36me3 peptide in the substrate competition experiment (Fig. 6c) supports a cis mechanism for demethylation by KDM4A and KDM4B.'

Because the above co-incubation competition assays allowed us to compare all three KDM4s, we have removed the original Fig. 6c from our manuscript, and now data for all three enzymes are included in the resubmission, as Fig. 6c and Supplementary Fig. 7d-e. Our current MS assays also allowed us to monitor the dimethyl and monomethyl products. The variability of background reaction rates with the previously included FDH-coupled demethylase assay made it difficult to obtain high-quality data amongst the KDM4s. The FDH-coupled demethylase assays are prone to high background rates, and until we are happy with the consistency of this assay, we prefer to leave these assays out of this publication.

Reviewer #3 (Remarks to the Author):

Su et al. identify and characterize the conserved double tudor domain (DTD) of KDM4 family members as a reader module for H3K23me3, a histone modification for which little is known. KDM4A, in addition to its known ability to bind H3K4me3 and H4K20me3, binds H3K23me3 while KDM4B exclusively binds H3K23me3 with high affinity. Structural analysis of KDM4A in complex with an H3K23me3 peptide revealed residues that are necessary for H3K23me3 interaction. By comparative analysis of KDM4A and KDM4B, the authors identified residues in KDM4B that provide exclusive binding preference for H3K23me3. To determine the functional role of KDM4B as an H3K23me3 reader, the authors identified mouse testes as a tissue with enriched levels of H3K23me3. KDM4B was determined to localize to regions of H3K23me3. Finally, binding of H3K23me3 by full-length KDM4B stimulated demethylation of H3K36me3 in vitro. The authors speculate that this mechanism is in place during meiosis to remove H3K36me3 and promote a heterochromatic state.

This manuscript is of interest, as little is known about H3K23me3, and the identification of the KDM4 family as readers of this mark provide potential mechanistic understanding of the role of this histone modification in chromatin organization. While the authors provide convincing evidence for the binding of H3K23me3 by KDM4 family members and comprehensive biochemical characterization of the intramolecular network that contributes to this interaction, several concerns should be addressed to support the major conclusions of the study.

Major concerns:

1. The authors use antibody capture followed by immunofluorescence to demonstrate antibody specificity for their in-house H3K23me3 antibody (Supplementary Figure 5b). However, only 3 additional histone marks (H3K27me3, H3K9me3, H3K4me3) were assayed to demonstrate specificity for H3K23me3. Given the conclusions drawn from the use of this antibody in Figure 5, more comprehensive specificity profiling of the H3K23me3 antibody needs to be done.

Response: This was the same concern of Reviewer 2, comment 1. We have now included additional data to support our conclusion. Please see our response to Reviewer 2.

2. (1) A major conclusion is that H3K23me3 is enriched in primary spermatocytes undergoing meiosis; however, only Figure 5c is presented in support of this conclusion, and the H3K23me3 antibody also detects this mark in spermatids. (2) This conclusion would be strengthened if a quantitative analysis of H3K23me3 among the different cell types was provided.

Response: (1) We thank the reviewer for this suggestion. The point we tried to convey was that H3K23me3 is only observed in cells that have started meiosis, aka meiotic and post-meiotic cells. Yes, new spermatids which have just completed meiosis II do contain somewhat reduced levels and more compact sub-nuclear distribution of H3K23me3, however, this PTM is essentially undetectable in mature sperm. To avoid any ambiguity with using 'undergoing meiosis', we have now changed the wording in the text (shown as follows).

(a) Abstract (Page 3):

'..., an interaction supported by in vivo co-localization of KDM4B and H3K23me3 at heterochromatin in mammalian **meiotic and newly post-meiotic** spermatocytes.'

(b) Highlights (Page 4):

'H3K23me3 is enriched in primary mammalian spermatocytes as they **start** meiosis.'

(c) Page 14 Subtitle:

H3K23 methylation is enriched in **primary** spermatocytes **that undergo meiosis**

(d) Page 15:

'We detected an enrichment of H3K23me3 in mouse primary spermatocytes as they began meiosis I, as indicated by the appearance of the meiosis-marker SCP3 (synaptonemal complex protein 3), which coincides with meiotic entry and leptotene. **In new spermatids which had completed meiosis II, H3K23me3 was also detected, albeit in a somewhat more compact sub-nuclear distribution than in spermatocytes; however, H3K23me3 was nearly undetectable in mature sperms (Supplementary Fig. 5d).'**

(2) The fact that these cells collectively form a tissue makes it extremely difficult to quantitate cellular amounts of H3K23me3 across individual cell types of testes/ seminal vesicles in a meaningful way. Therefore, to address Reviewer #3's comment and further support our conclusion that H3K23me3 increased once germ cells begin meiosis, we now provide more representative images in the Supplementary Fig. 5d.

3. (1) While the current data does support that H3K23me3 stimulates demethylation of H3K36me3 in vitro, the experiment presented is not able to discriminate between cis and trans interactions of KDM4B with these two histone modifications. Additional experiments need to be performed to support the claim of cis regulatory function. (2) In addition, demethylase assays should be done with a characterized mutant that disrupts H3K23me3 interaction.

Response: (1) We thank the reviewer for pointing this out. We agree with the

reviewer that using this peptide in our previous demethylation experiment (Fig. 6b) cannot unequivocally distinguish a cis or trans demethylation mechanism. Although our structural model (Supplementary Fig. 7a-b) and a new competition kinetic assay (Fig. 6c and Supplementary Fig. 7d) are in favor of the cis-histone mechanism (see below for more details), we reason a more in-depth kinetic analysis is needed to discriminate between cis and trans interactions, and is beyond the scope of this current study. We have now edited the text (see below) to avoid such confusion.

Page 6 (removed 'cis-histone'):

'We demonstrate that H3K23me3 binding stimulates ~~cis-histone~~ H3K36 demethylation by KDM4B.'

Page 17 (removed 'in-cis'):

'..., we reasoned that this interaction might enhance the activity of full-length KDM4B against certain histone methylations that co-occur with H3K23me3 ~~in-~~ ~~cis.~~'

Page 19 (added discussion about the new data of competition assay):

'The fact that the H3K23me3K36me3 peptide displayed an accelerated rate of demethylation relative to the K36me3 peptide in the substrate competition experiment (Fig. 6c and Supplementary Fig. 7d) supports a cis mechanism for demethylation by KDM4A and KDM4B.'

**Note about this piece of new data: In this competition assay, we incubated KDM4 enzyme with 1:1 ratio of H3K36me3 and H3K23me3K36me3 peptides. A cis mechanism would predict faster demethylation of the di-modified peptide, which is what we see for KDM4A and KDM4B (Fig. 6c and Supplementary Fig. 7d). A trans mechanism might predict similar rates of both peptides in the same solution. A trans mechanism in our system would reflect activation that allows K23me3 to bind the reader, and the K36me of either peptide to be more readily demethylated.*

Page 23 (added new discussion):

'Although our structural model and the competition kinetic assay suggest a cis-histone cross-talk mechanism between H3K23me3-binding and H3K27/K36 demethylation (Fig. 6c and Supplementary Fig. 7a-b), more detailed analysis will be needed to fully investigate cis-histone and trans-histone crosstalk between H3K23me3 and KDM4 demethylation sites (H3K9me, H3K36me and H3K27me), particularly using nucleosome substrates.'

(2) This was the same concern of Reviewer 2, comment 2 (2). We have now included additional data to support our conclusion. Please see our response to

Reviewer 2.

4. Cellular co-localization of KDM4B with H3K23me3 would be strengthened by assaying a DTD mutant unable to bind this mark.

Response: We thank the reviewer for this suggestion. The cellular co-localization of KDM4B and H3K23me3 (Fig. 5e) was performed on fixed rat testes tissue section. The suggested experiment could be done by introducing a point mutation into KDM4B in the wild-type mouse/rat, which will not be feasible in the given time frame. The alternative of using an H3K23me3-positive cell line is impossible at this point - we have assayed an array of mammalian culture cell lines (data not shown) for endogenous H3K23me3 expression and have not found one. We reason this is probably due to the specific expression pattern of H3K23me3 (enriched in meiotic and post-meiotic cells). Moreover, the enzymes catalyzing addition or removal of H3K23me3 have not been identified yet, so we can't manipulate H3K23me3 levels in an H3K23me3-negative cell line. Currently, very little is known about H3K23me3, which makes the suggested experiment very difficult. We believe the suggested experiment will be feasible in the future with more characterization of H3K23me3, but is beyond the scope of this current study.

To address the reviewer's concern (to strengthen the conclusion of co-localization between KDM4B and H3K23me3), we have performed the following set of experiments: (1) First, we generated the aromatic cage mutant of KDM4B-DTD (Y993A), and demonstrated its dramatic decrease of H3K23me3-binding ability by FP ($K_d = 327 \mu\text{M}$, Supplementary Fig. 4d). (2) We showed wild-type KDM4B-DTD but not the Y993A mutant can recognize H3Kc23me3 MLA histone in a far western blot (Supplementary Fig. 4e). (3) We also showed only the wild-type KDM4B-DTD but not the Y993A mutant can pull-down H3K23me3-containing MLA nucleosomes (Fig. 4d). We believe that our new data using more physiologically relevant chromatin substrate (nucleosomes) in far western blots and nucleosome pull-down, further supports our conclusion that KDM4B binds H3K23me3 via its reader domain.

Minor concerns:

1. For the fluorescence polarization (FP) experiments, only H3unmod (1-14) is listed as being used in FP. However, as the main focus of the paper is related to H3K23me3, a H3unmod (17-32) should also be used as control in binding experiments.

Response: We thank the reviewer for pointing this out. We reason it is not necessary to include the H3(17-32)unmod peptide for fluorescence polarization, since (1) we did not see obvious binding to H3K23unmod-containing peptides on peptide microarray, (2) H3(17-32)K23unmodK27me3 peptide only displayed

weak binding to the double tudor domains in FP assays (Fig. 1d), (3) we observed gradually decreasing binding with H3K23 methylation status changed from tri- to mono-methylation (Fig. 2e and Fig. 4a). The listed H3unmod (1-14) peptide was used as a control for H3K4me3 FP. To avoid any confusion, we have removed this peptide from Supplementary Table 1 and kept the biotin peptide pull-down with H3K4me3 (1-14) or H3K4me0 (1-14) peptide in Supplementary Fig. 1c in the revised manuscript.

2. For results provided on the histone peptide microarray, only the tri-methyl states for H3K23 are provided. Given the FP results with regard to di- and mono-methyl H3K23me3, it would be useful to the reader if the signals generated from these modifications were also provided. In addition, a more comprehensive analysis of the microarray data should be presented.

Response: We agree with the reviewer that such information will be useful for the readers. We have now followed the reviewer's suggestion and included a more comprehensive analysis of the microarray data in **Supplementary Fig. 1c**. This new analysis presents quantification of the signal intensities for different groups of peptides, including H3K23me3/2/1 (as suggested by the reviewer), H3K4me3/2, H4K20me3/2, H3K9me3, H3K14me3/2, H3K27me3/2, H3K36me3, H3K56me3, H3K79me3, and H3K115me3. Overall the result is consistent with the selective peptide spots in Fig. 1b.

3. In the methods section for the Mass Spectrometry-based demethylation assay, it indicates that this assay was performed with KDM4A-C; however, only the results from the KDM4B assay are presented. What were the results for KDM4A and KDM4C?

Response: We thank the reviewer for pointing this out. We have included a set of new experiments in which we incubated KDM4A-C enzymes with 1:1 mixed H3K36me3 and H3K23me3K36me3 substrates and monitored demethylation conversion over time ('co-incubation assay' in Fig. 6c and Supplementary Fig. 7d-e). The results are now described in the text as below.

Page 19:

'KDM4B preferentially demethylate H3K36me3 in the context of dually modified peptide when incubated with a 1:1 mixed pool of H3K23me3K36me3 and H3K36me3 peptide substrates (Fig. 6c). Consistent with the ability of both KDM4A and KDM4B but not KDM4C to bind H3K23me3 through its reader domain (Fig 1), demethylation of H3K36 was enhanced for KDM4A and KDM4B but not KDM4C (Supplementary Fig. 7d-e). The fact that the H3K23me3K36me3 peptide displayed an accelerated rate of demethylation relative to the K36me3 peptide in the substrate competition experiment (Fig. 6c) supports a cis mechanism for demethylation by KDM4A and KDM4B.'

Minor typos:

1. In Figure 2k, the Figure legend indicates that H3(19-27) is shown, but only H3(20-26) is shown in the figure.

Response: We thank the reviewer for this comment. We have corrected the figure legend (Page 46) accordingly.

2. In Figure 5A, the text states that H3K23me3 was profiled across adipose, testes, kidney, and liver. However, 2 lanes are labeled as testes and 0 lanes are labeled as liver.

Response: The labels on the figure were labeled correctly. We have corrected the figure legend (remove 'liver') on Page 46.

3. Some of the phrasing on pg. 18 is unclear:

Line 3,4,5: the authors state "we compared demethylation activity toward H3K27me3 or H3K36me3 by full-length KDM4B in the presence or absence of H3K23me3 on the same histone peptide (Figure 6b, Supplementary Fig. 6c)." However, Figure 6b shows that presence or absence of KDM4B with H3K23me3 in both experiments, while Supplementary Fig. 6c shows the demethylation assay with the absence of H3K23me3 in either the presence or absence of KDM4B. The sentence referring to these two different experiments needs to be clarified as it currently does not convey what the figures demonstrate.

Response: We have edited this section (Page 18) as follows:

'..., we compared demethylation activity toward H3K27me3 or H3K36me3 by full-length KDM4B in the presence ~~or absence~~ of H3K23me3 on the same histone peptide (Fig. 6b ~~and Supplementary Fig. 6c~~). Using tandem mass spectrometry, we did not detect significant demethylation of H3K27me3 peptide by KDM4B, even after prolonged incubation (Supplementary Fig. 6c), ...'

Line 9-11: the authors state: "The lack of measurable demethylation on the dually modified H3K23me3K27me3 peptide (Fig. 6b) indicated that the negligible demethylation of H3K27me3 was not be stimulated by H3K23me3 binding." This sentence needs clarification.

Response: We have edited this sentence (Page 18) as follows:

'The lack of measurable demethylation on the dually modified H3K23me3K27me3 peptide (Fig. 6b) indicated that ~~H3K23me3 binding by the~~

reader domain did not stimulate the negligible H3K27me3 demethylation.'

Line 22: the authors state: "depletion of substrate (H3K23me3)", but the substrate that is being depleted is H3K36me3.

Response: The reviewer is right about this typo. We have corrected this typo on Page 18 accordingly.

Reviewer #4 (Remarks to the Author):

This is a potentially interesting manuscript for Nature Communications. The biochemistry is the strength of the work. In regard to this I think it is essential to show the identified H3K23me3 tudor domain's alter regulate KDM4 KDM activity in the context of full length KDM/nucleosome interactions. Looking at the effect (or not) with respect to H3K9me3 activity (along with H3K36me3, the best characterised of the KDM4 activities) is essential. Specific experiments are outlined below. If this can be done (all possible) the next step is to demonstrate (via CHIP type analyses and appropriate materials) that the normal targeting mechanism is valid in cells (this could be in any organism from my perspective - yeast is OK). This would make a nice story.

Where I struggled with the 'story' was the connection of the biochemical work through to spermatogenesis/meiosis. As I read the paper the link to an H3K27/KDM4 tudor domain and KDM catalysis has not yet been made in vivo. The abstract is a bit misleading here with respect to "in vivo co-localisation"; this implies a direct interaction that has not actually been demonstrated.

I don't think it is necessary to make a full connection to physiology, but I think it needs to be made clear that this is, as yet, an undemonstrated possibility. I think aspects of the manuscript could be shortened, and it could be reorganised, separating biochemistry/structural biology, cellular interaction work and biological results. Some key primary experimental data, e.g. for the histone MS studies (which are non-trivial), is also missing and should be included.

Response: We thank Reviewer #4 for the very extensive comments and his/her appreciation of the biochemistry work we did in this paper.

Specific Points

Title: Possibly 'activity' or 'site of action' is better than 'functions'.

Response: We thank the reviewer for this suggestion. However, we reason 'function' in the original title would better summarize our observations in this study, which is not just about KDM4 enzyme 'activity' or 'site of action', but also about its connection with H3K23me3 in meiosis.

Abstract:

"...specific chromatin regions"?

Response: We have followed the reviewer's suggestion (see edits on Page 3).

Replace 'novel epigenetic mechanism' with 'a previously unidentified targeting

mechanism'. I am unsure if the role in spermatogenesis (and meiosis) has been sufficiently validated to mention in abstract - no need to over-claim.

Response: We have followed the reviewer's suggestion (see edits on Page 3).

P5. Demethylase subfamily.

Response: We have followed the reviewer's suggestion (see edits on Page 5).

Perhaps mention there are other KDM4s with different selectivity (KDM4D).

Response: We thank the reviewer for this suggestion. However, we think mentioning other KDM4s (such as KDM4D) in the introduction will be a distraction for the audience since the following study is focusing on KDM4A-C. We think it is better to include KDM4D in the discussion section (see edits on Page22):

'Interestingly, another KDM4 family member, KDM4D, which lacks the C-terminal reader domain and can only demethylate H3K9, has also been implicated in spermatogenesis⁴¹. This suggests other histone PTMs (like H3K9 methylation) might also be involved in germ cell development.'

The paragraph on 16 needs to be re-written - it presently reads as another abstract. This should set out the rationale of the work and key results.

Response: We thank the reviewer for this suggestion. However, we prefer to keep it relatively unchanged. The other three reviewers did not highlight this as an issue, so we hope reviewer 4 will be ok if we keep this original content.

The introduction could mention that the demethylation activity of some KDMs is known to be directed by methylation at non-demethylated sites, e.g. PHF8.

Response: We thank the reviewer for this comment. We did mention other KDMs with similar crosstalk between reader domain and catalytic domain in the discussion section (page 22 and cited below). Again, we tried to streamline the introduction, but do include the following in the discussion.

'These results are in accord with such "cross-talk" between a histone methylation "reader" and "eraser", leading to cis-histone demethylation reported for PHF8, KDM7A^{46,47}, KDM4A and KDM4C^{28,48}.'

P7. Express genes, not proteins.

Response: We have followed the reviewer's suggestion. The text on Page 7 is

now edited as

'We cloned and **expressed** human KDM4A-C double tudor domains (DTDs) as recombinant proteins with N-terminal affinity tags from *E. coli*'

P7. Mention binding assay and in the micro array.

Response: We thank the reviewer for this comment. We have mentioned the assays we used on Page 7 (listed as follows).

'... Analysis of the **peptide microarray assay** revealed

We further quantified the methylated histone binding of KDM4-DTDs by employing **a solution-based binding assay (fluorescence polarization**, Fig. 1c-d).
.....'

P7/8. I think it is important to support the fluorescence results with ITC or other established solution assays for binding (e.g. NMR).

Response: We thank the reviewer for this suggestion, though we do perform two types of binding assays. We believe fluorescence polarization is a well-established solution assay for binding. In our experience, binding affinity derived from FP was comparable to ITC (ITC quantification for KDM4A-DTD with H3K4me3 and H4K20me3 please refer to Garske, Craciun and Denu, *Biochemistry* 2008; and Garske, et al., *Nature Chemical Biology* 2010). In this manuscript, the binding affinity derived from FP was consistent with our peptide microarray analysis.

P8. It is difficult to distinguish between cation- π and hydrophobic interaction - see recent reported work on PHD domains.

Response: We agree with the reviewer on this comment. That is what we mean by

'The structure of the H3K23me3-bound KDM4A-DTD shows that the trimethyl-lysine group of H3K23 is coordinated **via cation- π and hydrophobic interactions** by an aromatic cage formed by F932, W967 and Y973 of the reader domain (Fig. 2d).' (Page 8)

P8. Electron density maps must be shown for all the complexed structures.

Response: We respectfully point out that we already showed electron density maps for all the complexed structures, which is the complexed structure for KDM4A-DTD with H3K23me3 peptide in Fig. 2b.

P10/11/12. I do appreciate that this is (in one aspect) principally a biochemical study, but I do think it is important to validate the relevance of the interactions

within (at least) the context of 'full length' KDM4 - nucleosome interactions. In this regard the statement on validation re biochemistry validating the interactions is misleading - it is biological validation that is important.

In this regard the results in Supp Fig 6a are crucial - I would separate these from the biochemical work and put in the main text. It is also essential to show the interaction is valid using full length KDM4 (OK to focus on one KDM4 and so more in depth assays rather than several KDM4s) and, appropriately, modified nucleosomes. The nucleosomes could be recombinant - selectively modified by enzymatic or chemical means, using wildtype and mutant KDM4s.

It is also important to test whether the potential new binding interactions influence the KDM activity of the KDM4s.

Response: We thank the reviewer for pointing out the importance of validating the interaction with nucleosome substrate. We have followed the reviewer's suggestion and moved the MARCC result to Main Figure 4c. To address the reviewer's concern (to strengthen the conclusion of co-localization between KDM4B and H3K23me3), we have performed the following set of experiments: (1) First, we generated the aromatic cage mutant of KDM4B-DTD (Y993A), and demonstrated its dramatic decrease of H3K23me3-binding ability by FP ($K_d = 327 \mu\text{M}$, Supplementary Fig. 4d). (2) We showed wild-type KDM4B-DTD but not the Y993A mutant can recognize H3K23me3 MLA histone in a far western blot (Supplementary Fig. 4e). (3) We also showed only the wild-type KDM4B-DTD but not the Y993A mutant can pull-down H3K23me3-containing MLA nucleosomes (Fig. 4d). We believe that our new data using more physiologically relevant chromatin substrate (nucleosomes) in far western blots and nucleosome pull-down further supports our conclusion that KDM4B binds H3K23me3 via its reader domain.

We realize there is a tremendous amount of detailed enzymology that will be necessary to provide a complete picture of the mechanisms utilized by KDM4A-C. We feel that such a comprehensive analysis is beyond the scope of this already comprehensive structure-function analysis. Examining KDM activity on nucleosomes with various combinatorial modifications is something we plan to complete in the next 1-2 years, but is beyond the scope of this current study.

P14. The work on the biological role of H3K23me3 is interesting. More detail, including primary data (including good quality MS/MS spectrum!) is required for the MS analysis. Such studies are non-trivial, especially to do quantitatively.

Response: We thank the reviewer for this suggestion. We have now included the MS/MS spectrum in Supplementary Fig. 6. We have previously published a more detailed description of our mass spec method in Reference 34 (Krautkramer, K.A., Reiter, L., Denu, J.M. & Dowell, J.A. Quantification of SAHA-Dependent Changes in Histone Modifications Using Data-Independent

Acquisition Mass Spectrometry. *J Proteome Res* 14, 3252-3262 (2015).).

P16. As noted above the potential unique ability (what is the evidence it is unique?) needs to be validated in more biologically relevant contexts. What is needed in the cellular analysis is not co-localisation but evidence for a direct interaction, using standard (but not necessarily easy) IP methods.

Response: We thank the reviewer for this suggestion. As the reviewer knows, capturing a relevant protein:protein interaction by IP experiments is difficult, and because the cells must be lysed to perform an IP, their relevance is often a concern. The best strategy would be designing in vivo FRET pairs, but unfortunately, this is beyond the scope of the current work. We believe that our biochemical evidence, the in vivo co-localization, and the various far-western and MARCC data provide a compelling case.

P17. The evidence for the co-occurrence of H3K23me3 and other markers needs to be defined - Fig.6a is a cartoon!

Response: We thank the reviewer for this comment. Fig. 6a was to illustrate the experimental design to look at crosstalk between H3K23me3 and other PTMs in vitro. Because of the novelty of H3K23me3, no one has looked into H3K23me3 co-occurrence with other markers in mammalian system. Previous mass spec analysis in *Tetrahymena* and *C. elegans* showed high co-occurrence of K23me3 and K27me3 (Methylation of histone H3K23 blocks DNA damage in pericentric heterochromatin during meiosis, 2014 *Elife*; Dynamic changes of histone H3 marks during *Caenorhabditis elegans* life cycle revealed by middle-down proteomics, 2016 *Proteomics*). Low co-occurrence between K23me3 and K9/36me3 was observed, which could be a result of either low abundance to begin with or active demethylation. To avoid such confusion as raised by the reviewer, we have now removed Fig. 6a.

P17. The 'modelling' paragraph could be cut/moved to Discussion. Easiest just to report the experimental evidence.

Response: We thank the reviewer for this suggestion. We only included the models in Supplementary Figures. We reason it is necessary to keep corresponding text in the results section to help guide the flow of the paper (in regards to why we selected K27 and K36 as potential demethylation site).

P18. The results that H3K36me3 demethylation is enhanced by H3K23 methylation resulting from a reduction in Km is interesting.

(1) It is also important to test for an effect on the other established KDM4 substrate, i.e. H3K9me3/2. (2) Also important to demonstrated that the increased activity is due to the K3K23/KDM4 interaction by appropriate mutagenesis

studies.

Response: (1) We thank the reviewer for this suggestion. We agree with the reviewer that K9 will also be an interesting target to check. But based on our structural modeling, K27 and K36 are more likely to be the targets. Nevertheless, we have now modified the discussion to include the possibility of K9 being a demethylation target while binding K23 (see below).

Page 22:

'Although our structural model suggests cis-histone cross-talk mechanism between H3K23me3-binding and H3K27/K36 demethylation (Supplementary Fig. 7a-b), more detailed analysis is still needed to fully access cis-histone and trans-histone crosstalk between H3K23me3 and KDM4 demethylation sites (H3K9me, H3K36me and H3K27me) using nucleosome substrates.'

(2) We thank the reviewer for this suggestion. Please see our response to Reviewer 2 comment 2.

Discussion

I would focus on biochemical results first - as indicated above with discussing in the context of preliminary work that catalytic KDM activity is targeted by binding to non-catalytic domains.

Response: We thank the reviewer for this suggestion. We agree with the reviewer to focus on biochemical results first. In discussion, we first discuss the biochemical results about KDM4 reader domain binding. Then we talk about the H3K23me3 and KDM crosstalk in the context of germ cell development, among which we discuss the catalytic domain crosstalk on Page 23.

Overall, although the results on H3K23 in meiosis / spermatogenesis are interesting, I am not sure they are sufficiently well connected to the biochemical results. If this connection is to be made, at least, good quality evidence for the targeting of KDM4 KDM activity by H3K27 methylation in cells needs to be presented.

Response: We thank the reviewer for this suggestion. However, such experiment is impossible to do at this point. We have only identified H3K23me3 in testes tissues, not other tissue or any cell lines. We reason this is probably due to the specific expression pattern of H3K23me3 (enriched in meiotic and post-meiotic cells). Moreover, the enzymes catalyzing addition or removal of H3K23me3 have not been identified yet, so we can't manipulate H3K23me3 levels in an H3K23me3-negative cell line. Currently, very little is known about H3K23me3, which makes the suggested experiment very difficult. We believe the suggested experiment will be feasible in the future with more characterization of

H3K23me3, but is beyond the scope of current study.

Reviewer #1 (Remarks to the Author):

The revised manuscript is now acceptable for publication.

Reviewer #2 (Remarks to the Author):

The authors have addressed my concerns satisfactorily.

Reviewer #3 (Remarks to the Author):

Su et al. have provided an improved manuscript that is suitable for publication in Nature Communications. The authors have provided the requested experiment demonstrating the specificity of the H3K23me3 antibody. The authors have sufficiently addressed the concern regarding enrichment of H3K23me3 in spermatocytes and spermatids by clarifying the language and providing additional representative images. The cis-histone conclusion from the previous manuscript has been addressed by changing the language and providing a new competition assay. Finally, the authors have provided rationale and additional experiments to demonstrate that Y993A mutation decreases KDM4B's ability to bind H3K23me3. All minor concerns have also been addressed.

Reviewer #4 (Remarks to the Author):

As before I think the strength of this manuscript is the biochemistry and structural biology at the very least raising the possibility that H3K23 methylation is biologically important. These aspects of the work have been further strengthened in the revised manuscript. Nonetheless the major gap in the work remains the unequivocal demonstration that KDM4B-DTD interacts with H3K23me3 in a functionally relevant manner concerning spermatogenesis meiosis (and how selective the effect is to spermatogenesis) inside cells. The biochemical work is consistent with this, but, I think, not more. However, I don't think this should preclude publication of what I think is an interesting study on H3K23 methylation, only that the claims for functional relevance to biology should be toned down. I agree

with reviewer 1, the biological mechanism is plausible, which in my view is enough – there is no need to claim beyond this (e.g. as the last two sentences of the abstract do). I'd advise the authors to tone down the biological function claims.

Overall I'm in favour of publication but think there should be further modifications (though appreciate the other reviewers may disagree). The following comments are intended to be helpful with respect to improving the manuscript - new experiments are not necessarily needed (possibly excepting looking at the effect of K23 methylation on K9 demethylation,), unless the authors wish to provide clear evidence for biologically relevant function.

Specific comments

With respect to reviewers 2 / 3

The antibody characterisation response is good– this is much more than most in the field do.

Also the biochemical cage mutant study is nice.

Some recombinant full length KDM4 proteins have in fact been made in human cells, but using them in the type of binding studies required for the current work could be tricky so I agree beyond current scope.

Page 19

Is actual data shown for Fig 6c? The effect on increasing demethylation is potentially interesting, but is not as great as, e.g. that observed for stimulation (or inhibition) with PHF8/KIA1718 by H3K4methylation. I see no need to overclaim re functional relevance. Clearly also further kinetic / binding data would be useful – but this could be subject of a further study providing the observation is presented as preliminary and of potential biological relevance.

Note that KDM activity is often affected by substrate / nature length so the relatively small differences observed may not be present with different substrates / conditions.

Figure 6c – how many repeats were done (n=?). I hope the errors are ‘statistical’ arising from analysis of different charge states as the legend seems to indicate.

What about looking at effect of K23 methylation on demethylation at K9?

I’d avoid use of ‘cis’ in relation to the enhanced activity (in cells its possible different histones are involved). Important to tone down claims re functional relevance of preliminary kinetic data including in discussion.

The new data on colocalisation in response to reviewer 3 is all biochemical, though the addition of data with nucleosomes is good (please cite method for preparation).

With respect to reviewer 4

Many thanks for work to address the issue raised by me.

Re title – I don’t think the evidence is presented to make the claim inherent in the title, but others may disagree.

Re repeat of abstract in introduction – editorial decision I think its unnecessary.

Re cross talk issue and PHF8/KIA 1718 etc, I’d rewrite this – in the case of PHF8 there is very strong evidence H3K4methylation strongly promotes demethylation at K9 including by crystallography – why not just simply say this - it's a good precedent.

FP is not the same as ITC – it is a complementary technique giving different information, but I accept may be beyond current scope.

RE pi- cation I would say “H3K23 is bound via cation- π and / or hydrophobic interactions...”

Re maps – apologies my mistake, I meant representative maps for all structures (in supp info) not just complexed one.

Re the issue of demonstrating the interaction in cells under “natural circumstances”, I agree this is often difficult, but it’s absolutely crucial to the biological relevance of the work. As indicated above the biochemical evidence is consistent with a cellular interaction, but, I think, no more (100s of proteins will probably bind to / act on histone tails in non biologically relevant manners).

Reviewer #1 (Remarks to the Author):

The revised manuscript is now acceptable for publication.

Reviewer #2 (Remarks to the Author):

The authors have addressed my concerns satisfactorily.

Reviewer #3 (Remarks to the Author):

Su et al. have provided an improved manuscript that is suitable for publication in Nature Communications. The authors have provided the requested experiment demonstrating the specificity of the H3K23me3 antibody. The authors have sufficiently addressed the concern regarding enrichment of H3K23me3 in spermatocytes and spermatids by clarifying the language and providing additional representative images. The cis-histone conclusion from the previous manuscript has been addressed by changing the language and providing a new competition assay. Finally, the authors have provided rationale and additional experiments to demonstrate that Y993A mutation decreases KDM4B's ability to bind H3K23me3. All minor concerns have also been addressed.

Reviewer #4 (Remarks to the Author):

As before I think the strength of this manuscript is the biochemistry and structural biology at the very least raising the possibility that H3K23 methylation is biologically important. These aspects of the work have been further strengthened in the revised manuscript. Nonetheless the major gap in the work remains the unequivocal demonstration that KDM4B-DTD interacts with H3K23me3 in a functionally relevant manner concerning spermatogenesis meiosis (and how selective the effect is to spermatogenesis) inside cells. The biochemical work is consistent with this, but, I think, not more. However, I don't think this should preclude publication of what I think is an interesting study on H3K23 methylation, only that the claims for functional relevance to biology should be toned down. I agree with reviewer 1, the biological mechanism is plausible, which in my view is enough – there is no need to claim beyond this (e.g. as the last two sentences of the abstract do). I'd advise the authors to tone down the biological function claims.

Overall I'm in favour of publication but think there should be further modifications (though appreciate the other reviewers may disagree). The following comments are intended to be helpful with respect to improving the manuscript - new experiments are not necessarily needed (possibly excepting looking at the effect of K23 methylation on K9 demethylation,), unless the authors wish to provide

clear evidence for biologically relevant function.

Response: We thank reviewer 4 for this suggestion. We have now toned down the last two sentences in abstract (Page 3) as follows:

“In vitro demethylation assays suggest H3K23me3 binding by KDM4B stimulates H3K36 demethylation. Together, these results provide a possible mechanism whereby H3K23me3-binding by KDM4B directs localized H3K36 demethylation during meiosis and spermatogenesis.”

Specific comments

With respect to reviewers 2 / 3

The antibody characterisation response is good– this is much more than most in the field do.

Also the biochemical cage mutant study is nice.

Some recombinant full length KDM4 proteins have in fact been made in human cells, but using them in the type of binding studies required for the current work could be tricky so I agree beyond current scope.

Page 19

Is actual data shown for Fig 6c? The effect on increasing demethylation is potentially interesting, but is not as great as, e.g. that observed for stimulation (or inhibition) with PHF8/KIA1718 by H3K4methylation. **I see no need to overclaim re functional relevance.** Clearly also further kinetic / binding data would be useful – but this could be subject of a further study providing the observation is presented as preliminary and of potential biological relevance.

Note that KDM activity is often affected by substrate / nature length so the relatively small differences observed may not be present with different substrates / conditions.

Response: Fig 6c shows quantification from actual data obtained via mass spec analysis. Yes we agree the effect is not as great as PHF8/KIA1718.

Figure 6c – how many repeats were done (n =?). I hope the errors are ‘statistical’ arising from analysis of different charge states as the legend seems to indicate.

Response: We have performed this analysis for KDM4A-C with varying peptide

concentration and varying time points (data not shown) and observed similar trend as we presented here. The statistical analysis (student's t test, $p < 0.01$) is now described in the figure legend.

What about looking at effect of K23 methylation on demethylation at K9?

Response: We appreciate the reviewer's interest in this. We reason this is beyond the scope of this manuscript and will be a good topic for follow-up study.

I'd avoid use of 'cis' in relation to the enhanced activity (in cells its possible different histones are involved). Important to tone down claims re functional relevance of preliminary kinetic data including in discussion.

Response: We have further toned down the "cis" activity by removing this statement in abstract and modified the discussion as follows (page 23):
"Although our structural model and the preliminary competition kinetic assay suggests a cis-histone cross-talk mechanism between H3K23me3-binding and H3K27/K36 demethylation (Fig. 6c and Supplementary Fig. 7a-b), more detailed kinetic analysis will be needed to fully investigate cis-histone and trans-histone crosstalk between H3K23me3 and KDM4 demethylation sites (H3K9me, H3K36me and H3K27me), particularly using nucleosome substrates."

The new data on colocalisation in response to reviewer 3 is all biochemical, though the addition of data with nucleosomes is good (please cite method for preparation).

Response: We have now cited the method for preparation in corresponding methods section (page29-30).

With respect to reviewer 4

Many thanks for work to address the issue raised by me.

Re title –I don't think the evidence is presented to make the claim inherent in the title, but others may disagree.

Re repeat of abstract in introduction – editorial decision I think its unnecessary.

Response: We leave this for the editor to decide.

Re cross talk issue and PHF8/KIA 1718 etc, I'd rewrite this – in the case of PHF8

there is very strong evidence H3K4methylation strongly promotes demethylation at K9 including by crystallography – why not just simply say this - it's a good precedent.

Response: We have now added in such discussion on Page23:

“In the case of PHF8, biochemical and structural characterization indicates H3K4me3 strongly promotes H3K9 demethylation⁴⁶.”

FP is not the same as ITC – it is a complementary technique giving different information, but I accept may be beyond current scope.

RE pi- cation I would say “H3K23 is bound via cation- π and / or hydrophobic interactions...”

Response: We have now edited the text according to this reviewer’s suggestion (page 8).

Re maps – apologies my mistake, I meant representative maps for all structures (in supp info) not just complexed one.

Response: We have deposited all structures into PDB and the maps could be retrieved more easily through the database.

Re the issue of demonstrating the interaction in cells under “natural circumstances”, I agree this is often difficult, but it’s absolutely crucial to the biological relevance of the work. As indicated above the biochemical evidence is consistent with a cellular interaction, but, I think, no more (100s of proteins will probably bind to / act on histone tails in non biologically relevant manners).

Response: That might be true, but KDM4 is not one of them. This is more of a philosophical argument about the inherent shortcomings of in vivo vs. in vitro evidence of direct interactions. We would argue that our extensive biochemical and structural characterization of the KDM4-K23me3 provides extremely compelling evidence that this interaction is highly specific. Our very first experiment demonstrated that among ~1000 histone peptides, the KDM4 family binds with great specificity in aa sequence and in a methylation dependent manner. We are not aware of any in vivo experiment that provides such unequivocal results that would say protein X directly interacts with Y. FRET, crosslinking, IP, etc., produce their own caveats.